# NavOL: Navigation Policy with Online Imitation Learning

**Xiaofei Wei** [* 1 2]  **Chun Gu** [* 1 2]  **Li Zhang** [1 2]

`logosroboticsgroup.github.io/NavOL`

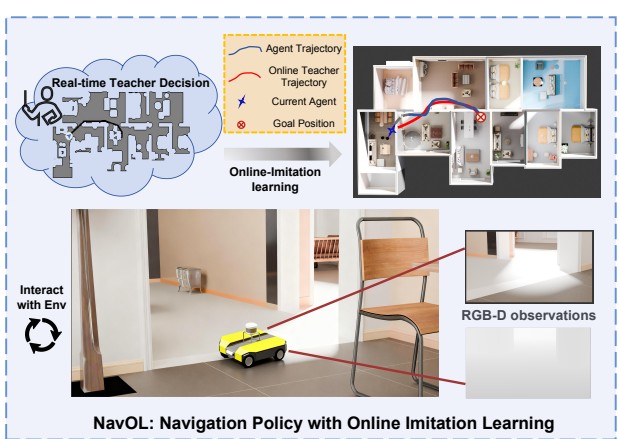
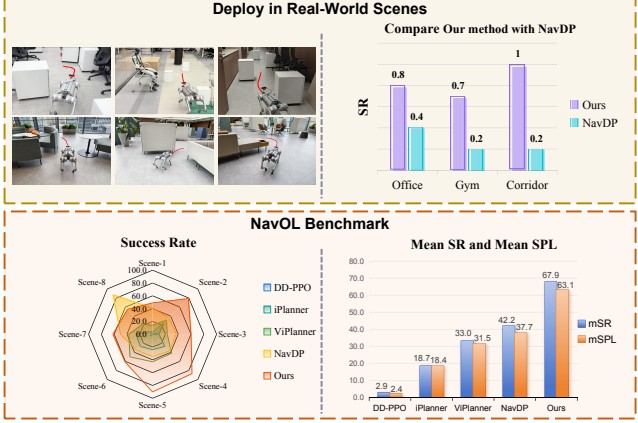

*Figure 1. (Left)* Online imitation learning framework: the agent interacts with the environment, taking ego-view RGB-D observations and the target goal as input, while a global planner provides online expert trajectories for guidance. The policy learns online by imitating the expert's decisions, forming a closed interaction loop within the IsaacLab simulator. *(Right)* Evaluation on a new benchmark built from 3D-Front Dataset (Fu et al., 2021). *NavOL* achieves the best performance in terms of success rate (SR) and success weighted by path length (SPL), showcasing its superior robustness and generalization.

## Abstract

Learning robust navigation policies remains a core challenge in robotics. Offline imitation learning suffers from distribution shift and compounding errors at rollout, while reinforcement learning requires reward engineering and learns inefficiently. In this paper, we propose *NavOL*, an online imitation learning paradigm that interacts with a simulator and updates itself using expert demonstrations gathered online. Built upon a pretrained navigation diffusion policy that maps local observations to future waypoints, *NavOL* trains in a rollout–update loop: during rollout, the policy acts in the simulator and queries a global planner which has privileged access to the global environment for the optimal path segment as ground truth trajectory labels; during update,

the policy is trained on the online collected observation–trajectory pairs. This online imitation loop removes the need for reward design, improves learning efficiency, and mitigates distribution shift by training on the policy's own explored rollouts. Built on IsaacLab with fast, high-fidelity parallel rendering and domain randomization of camera pose and start-goal pairs, our system scales across 50 scenes on 8 RTX 4090 GPUs, collecting over 2,000 new trajectories per hour, each averaging more than 400 steps. We also introduce an indoor visual navigation benchmark with predefined start and goal positions for zero-shot generalization. Extensive evaluations on simulation benchmarks, including the NavDP benchmark and our proposed benchmark, as well as carefully designed real-world experiments, demonstrate the effectiveness of *NavOL*, showing consistent performance gains in online imitation learning.

---

[*]Equal contribution [1]School of Data Science, Fudan University [2]Shanghai Innovation Institute. Correspondence to: Li Zhang <lizhangfd@fudan.edu.cn>.

*Proceedings of the 43rd International Conference on Machine Learning*, Seoul, South Korea. PMLR 306, 2026. Copyright 2026 by the author(s).

## 1. Introduction

Open-world embodied navigation (Anderson et al., 2018; Ku et al., 2020) arises from the human desire for robots to

autonomously navigate through unfamiliar environments by observing their surroundings and reaching specified targets. The environments are dynamic, partially observable, and cluttered, requiring agents to translate perception and goal conditions into executable action control to achieve goals safely and efficiently. The field has progressed from traditional discrete mapping and planning (Chaplot et al., 2020; Campos et al., 2021; Labbé & Michaud, 2018) toward approaches that integrate more expressive visual representations (Khandelwal et al., 2022; Majumdar et al., 2023). However, a central bottleneck remains the scarcity of large-scale, high-quality training data. While some studies (Hirose et al., 2019; Karnan et al., 2022; Hirose et al., 2023) collect real-world trajectories to supervise policy learning, large-scale collection and annotation are costly, constraining model capacity and robustness. In contrast, another line of works (Cai et al., 2025; Contributors, 2025) show that large-scale synthetic data and high-fidelity 3D assets can substantially enhance sim-to-real transfer and cross-embodiment policy learning, highlighting the potential of a more scalable and effective path forward: exploiting the realism and diversity of synthetic data, along with its seamless integration with modern high-fidelity simulators (Todorov et al., 2012; Szot et al., 2021; NVIDIA).

Substantial efforts (Meng et al., 2025; Sridhar et al., 2024; Cai et al., 2025; Shi et al., 2025) have been made to leverage demonstration trajectories for imitation learning. NavDP (Cai et al., 2025) implements a data generation pipeline that automatically produces navigation trajectories in simulation using public 3D scene assets (Fu et al., 2021; Chang et al., 2017), achieving over 20× higher efficiency compared to real-world data collection. With this large-scale dataset, NavDP demonstrates strong zero-shot generalization across different robot embodiments. However, despite its use of a critic head to select the best sampled trajectories, its capability remains constrained by the quality and diversity of the pre-collected demonstration data, as it only learns to replicate recorded behaviors without interacting with the environment. When deploying, the policy tends to drift out of the training distribution due to compounding errors during rollout, leading to unstable navigation. In contrast, reinforcement learning (Zeng et al., 2024; Eftekhar et al., 2024; Zhu et al., 2025) enables agents to improve through direct interaction with the environment. Although this approach enables continual policy refinement beyond demonstration data, it heavily depends on carefully designed reward functions and often suffers from sparse rewards, resulting in low learning efficiency and making large-scale training in complex environments challenging.

To address these limitations, we combine the sample efficiency of imitation learning with the interactive benefits of online learning. In this paper, we introduce a novel online imitation learning paradigm for visual navigation, *NavOL*,

that interacts with a simulator and updates itself iteratively using expert demonstrations gathered online. *NavOL* is initialized from a pretrained navigation diffusion policy (Cai et al., 2025), which maps RGB and depth observations to future waypoints and provides a strong initial behavioral prior for rollouts. While DAgger-style algorithms (Ross et al., 2011) address distribution shift, scaling them to high-dimensional visual diffusion policies has remained computationally prohibitive. This limitation often restricts previous works to offline data synthesis, thereby failing to incorporate the real-time interaction necessary to correct distribution drift (Zhang et al., 2024). Building on DAgger algorithm, we propose *NavOL*, which leverages massive parallel simulation to bridge this gap. Unlike traditional DAgger applied to simple controllers, we demonstrate how to stabilize the online fine-tuning of generative diffusion policies. Our method follows a rollout-update loop: during rollout, the policy navigates in the simulator and, at each step, a global path planner computes the optimal path segment from the current pose to the goal. This trajectory segment serves as supervision for the current observation. During update, we train the diffusion policy with the standard denoising objective used in DDPM (Ho et al., 2020) on the collected observation–trajectory pairs. This online imitation loop avoids reward engineering and the low sample efficiency of reinforcement learning, while also reducing distribution shift and compounding errors in offline imitation learning by training on the policy's explored rollouts. With careful implementation built on IsaacLab's (Mittal et al., 2023) high-performance, high-fidelity rendering, we run a large number of environments in parallel and train across 50 3D scene assets. With only 8 RTX 4090 GPUs, the system collects more than 2,000 new high-quality trajectories per hour, each averaging over 400 steps, enabling rapid online improvement at scale. We further apply domain randomization to camera pose and lighting to increase diversity.

Extensive experiments on the NavDP benchmark and our newly introduced benchmark show that *NavOL* consistently outperforms competing methods across all metrics. Furthermore, to demonstrate the sim-to-real capability and cross-embodiment capability of *NavOL*, we conduct real-world experiments on carefully designed scenarios. As shown in Figure 1, *NavOL* achieves strong performance and surpasses competitors by a large margin, demonstrating that incorporating online planner supervision into imitation learning significantly enhances navigation performance.

The contributions of this paper are as follows: **(1)** We present *NavOL*, an online imitation learning framework for navigation policy learning. **(2)** We implement a highly parallelized rollout–update pipeline that, on 8 RTX 4090 GPUs, collects and learns from over 2,000 new trajectories in 50 scenes per hour. **(3)** We introduce a visual indoor navigation benchmark for comprehensive evaluation. **(4)** We conduct

extensive experiments on both simulation benchmarks and real-world scenarios, showing that *NavOL* delivers consistent performance gains in online imitation learning.

## 2. Related work

### 2.1. Vision navigation

Robot navigation focuses on efficiently accomplishing advanced navigation tasks in unfamiliar environments using robotic systems equipped with egocentric sensors. Classical vision-based navigation systems (Labbé & Michaud, 2018; Konolige, 2000; Fox et al., 2002) decompose the problem into modular sub-tasks solved by specialized components, but struggle with high-dimensional sensory inputs and complex environments. Recent work has therefore shifted toward end-to-end learning-based approaches. Unsupervised methods such as iPlanner (Yang et al., 2023) and ViPlanner (Roth et al., 2024) rely on hand-designed cost-map supervision and simplified dynamics, which makes them prone to local minima and limits generalization and sim-to-real transfer. Reinforcement learning approaches, enabled by large-scale simulators (Puig et al., 2023; Xiang et al., 2020; Salimpour et al., 2025; Makoviychuk et al., 2021; Mittal et al., 2023), have shown strong performance but often suffer from large search spaces, sparse rewards, and heavy reliance on reward shaping and auxiliary losses (Ye et al., 2021; Singh et al., 2023; Wang et al., 2019). Imitation learning (Shah et al., 2022; Cai et al., 2025; Ding et al., 2019) offers a simpler alternative by learning directly from expert demonstrations, but offline settings are limited by dataset coverage and suffer from covariate shift and compounding errors. In contrast, we adopt an online imitation learning framework that rolls out policies in simulation, queries a global planner for stepwise waypoint supervision, and trains on the policy's own explored rollouts, and scaling efficiently in IsaacLab (Mittal et al., 2023).

### 2.2. Diffusion models in robotics

Diffusion models (Ho et al., 2020) offer stable training and strong capacity to model multimodal distributions, and have shown substantial promise for robot policy learning. Diffusion-based policies have been applied across robotics, for visuomotor manipulation (Chi et al., 2023), trajectory generation for navigation (Sridhar et al., 2024), and multi-skill locomotion (Huang et al., 2024). In visual navigation, ViNT (Shah et al., 2023) uses diffusion to propose diverse exploration subgoals and predict future images for long-horizon planning, while NoMaD (Sridhar et al., 2024) directly infers multimodal actions from visual observations. However, these policies are typically trained offline on large teleoperation datasets, which are expensive to collect and hard to scale, limiting real-world performance. To bypass this bottleneck, NavDP (Cai et al.,

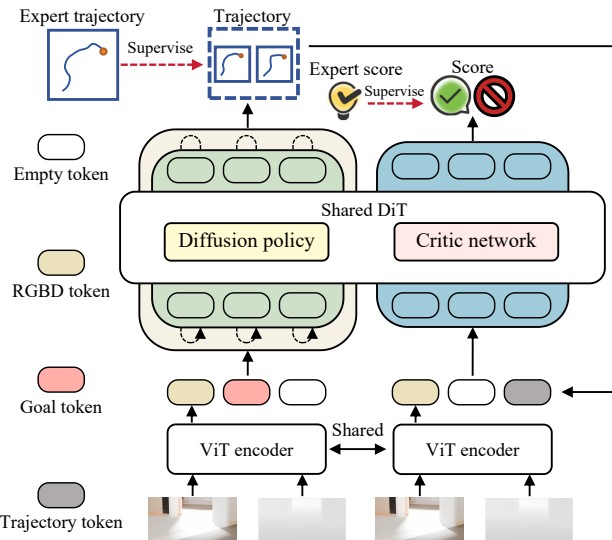

*Figure 2.* An overview of the *NavOL* model architecture.

2025) synthesizes large navigation datasets entirely in simulation using public 3D assets (Fu et al., 2021; Chang et al., 2017), and demonstrates zero-shot sim-to-real transfer and cross-embodiment generalization with simulation-only training. However, NavDP (Cai et al., 2025) remains an offline imitation pipeline over a fixed corpus and thus inherits limitations: behavior bounded by demonstration coverage and diversity, no interaction during training, and covariate shift with compounding errors at rollout. Building on these insights, we move from offline to online by training a diffusion-based navigation policy in the simulator via an online rollout–update loop, reducing distribution shift and compounding errors while scaling efficiently with efficient implementation built upon IsaacLab (Mittal et al., 2023) over a variety of scenes.

## 3. Method

In this section, we introduce *NavOL*, an online imitation learning framework for navigation. We describe the model architecture (Sec. 3.1), the online imitation learning pipeline (Sec. 3.2), and the scene setup and indoor navigation benchmark (Sec. 3.3). An overview is shown in Figure 3.

### 3.1. Model architecture

As shown in Figure 2, *NavOL* builds on a multimodal diffusion policy (Cai et al., 2025) and comprises two cooperating components: a trajectory generator $f$ that proposes candidate waypoint sequences and a critic that scores their safety and feasibility before execution. Given RGB–D observations $o$ and an optional goal instruction $g$ (point, image, pixel, or no-goal), the generator predicts a short-horizon sequence of relative waypoints:

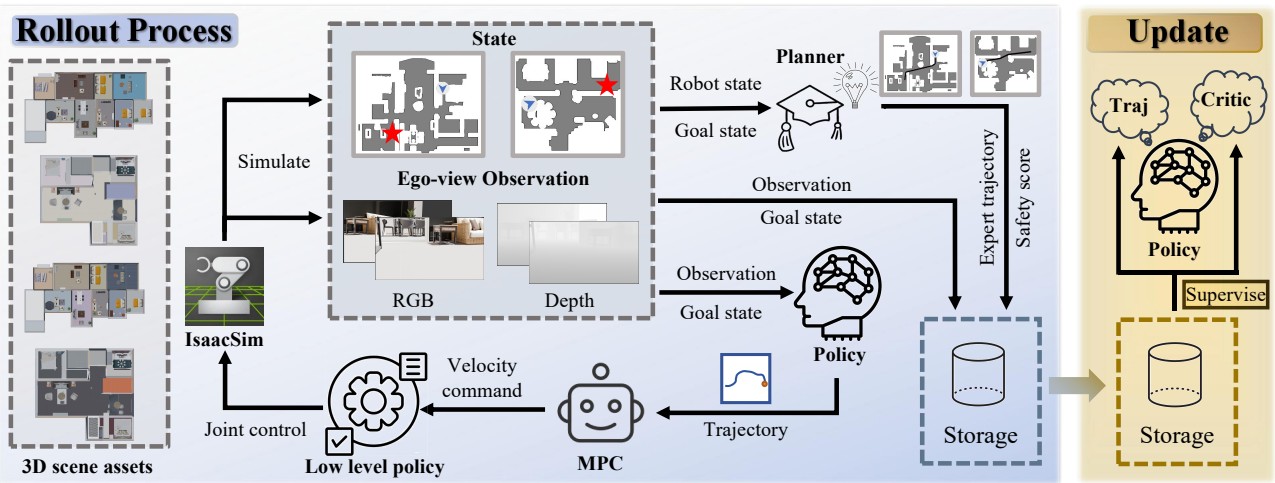

*Figure 3.* The overall pipeline of *NavOL* alternates between a rollout phase and an update phase, forming a closed interaction loop within the IsaacSim. In the rollout stage, the agent navigates within IsaacSim, taking ego-view RGB-D observations and the target goal as input. Then, the diffusion policy outputs waypoint trajectories, which are tracked by Model Predictive Control (MPC) and a low-level controller to produce the agent's physical actions. Concurrently, a global planner with privileged access to the environment provides expert trajectories and safety scores, which serve as supervision signals. The collected data are stored in a storage for jointly optimizing the trajectory generation and the critic network in the update stage.

$$f(o, g) = \{(\Delta x_i, \Delta y_i, \Delta \omega_i)\}_i, \qquad (1)$$

where $\Delta x_i$ and $\Delta y_i$ represent the relative distance of waypoints in the robot centric frame, and $\Delta \omega_i$ denotes the heading change in radians. Specifically, we encode the RGB and depth streams with a DepthAnythingV2 (Yang et al., 2024) ViT backbones, and a transformer decoder compresses the features into 16 fused tokens. Goal instructions are embedded (e.g. an MLP for point goals) and then combined with the fused visual tokens to define the conditioning context. Conditioned on these tokens, a Diffusion Transformer (DiT) with a DDPM scheduler (Ho et al., 2020) denoises from Gaussian noise to a $K$-step waypoint sequence via cross-attention to the condition embeddings. The critic $V$ shares the visual tokens and DiT backbone with the generator $f$ for consistent spatial features. A candidate trajectory $a$ is encoded into a trajectory token and, together with the fused RGB–D tokens (without goal tokens to remain goal-agnostic), is processed by the shared DiT to produce a scalar safety score $v = V(o, a)$ via a lightweight head. In deployment, we rank the trajectories sampled by the diffusion policy using this score and execute the top-ranked plan.

### 3.2. Online imitation learning

**Expert planner.** Our framework relies on a global planner with privileged access to the environment map, which is unavailable during real-world deployment, to provide real-time supervision during training. We implement the planner on the navigation mesh (NavMesh) from Habitat-Sim (Puig et al., 2023), which encodes walkable surfaces under agent-

specific parameters (e.g., height and radius). The planner is used to compute globally optimal paths during rollouts and to sample start–goal pairs for episode initialization.

Given the agent pose $p$ and goal position $g$, the planner $f'$ queries the NavMesh to obtain a shortest path represented by sparse waypoints. To make the path suitable for control, we densify and smooth it through post-processing: we adjust waypoints to maximize clearance from obstacles by performing a line search in the local area, resample the path at uniform arc-length intervals, fit a 2D cubic spline for smoothness, and project the resulting points back onto the NavMesh to ensure collision-free trajectories. The final planned trajectory is expressed as:

$$f'(p, g) = \{(\Delta x_i', \Delta y_i', \Delta \omega_i')\}_i, \qquad (2)$$

where $\Delta x_i'$, $\Delta y_i'$, $\Delta \omega_i'$ are relative waypoints transformed from the planned trajectory.

To increase task diversity and difficulty, we repeatedly sample start–goal pairs uniformly over the navigable surface until the resulting raw path contains more than $F$ keypoints, indicating at least $F$ heading changes. This criterion favors more challenging scenes with richer obstacle interactions, yielding higher-quality training episodes. Notably, querying expert trajectories is computationally efficient and introduces negligible overhead during training, without affecting overall training speed.

**Online data aggregation.** To address the distribution shift of offline imitation learning, inspired by DAgger (Ross et al., 2011), we split each training iteration into a $T$-step rollout

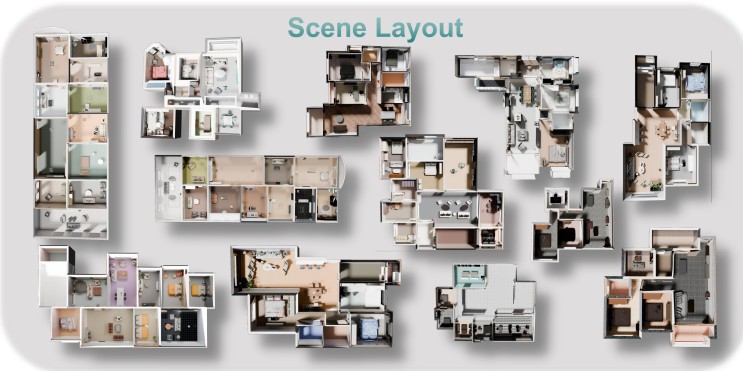
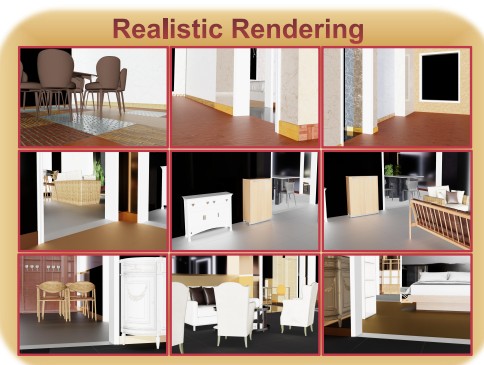

*Figure 4.* An overview of our proposed benchmark. The left section presents the top-down layouts of scene assets, while the right section showcases the photorealistic renderings within the IsaacLab simulator.

stage and an update stage. During rollout, the policy is trained on the states it actually visits, mitigating covariate shift and compounding errors.

We use IsaacLab (Makoviychuk et al., 2021) for GPU-parallel simulation. At each step $t$, the agent receives RGB-D observations $o_t$, which are encoded into visual tokens (Section 3.1). Conditioning on these tokens and the goal $\boldsymbol{g}_t$, *NavOL* outputs waypoint actions $\boldsymbol{a}_t^{\text{policy}} = f(\boldsymbol{o}_t, \boldsymbol{g}_t)$. In parallel, the expert planner provides optimal waypoints $\boldsymbol{a}_t^* = f'(\boldsymbol{p}_t, \boldsymbol{g}_t)$. The executed action is sampled as

$$\boldsymbol{a}_t = \begin{cases} \boldsymbol{a}_t^{\text{policy}}, & \text{with probability } \rho, \\ \boldsymbol{a}_t^*, & \text{with probability } 1 - \rho. \end{cases} \quad (3)$$

Occasionally executing expert actions stabilizes training and improves performance. Waypoints are converted to velocity commands via MPC and then to joint controls by a low-level controller. Unlike prior offline rendering approaches (Cai et al., 2025), our fully embodied online rollouts naturally capture motion-induced visual effects, such as camera shake, improving generalization.

Using the navigation mesh, we compute the distance $d_t^i$ from each waypoint to its nearest obstacle and derive a target safety score to supervise the critic:

$$v_t' = -\sum_{i=0}^{K} \mathbb{I}(d_t^i < d_{\text{safe}}) + \alpha \sum_{i=0}^{K-1} (d_t^{i+1} - d_t^i) \quad (4)$$

where $d_{\text{safe}}$ is a safety threshold, and $\alpha$ is a re-weight hyperparameter. From this online interactive data aggregation, we collect the data tuples $\{(\boldsymbol{o}_t, \boldsymbol{g}_t, \boldsymbol{a}_t^*, v_t')\}_{t=1\ldots\text{T}}$ which constitute the training data for the subsequent update stage.

**Policy optimization.** We train the navigation diffusion policy and critic using data collected from the most recent rollout stage, discarding trajectories from previous iterations and retaining only the current rollout data to maximize the

efficiency of network updates. The policy is optimized with a standard denoising objective conditioned on observations $o$, goals $g$, and expert trajectories $a$:

$$\mathcal{L}^{\text{actor}} = \text{MSE}(\epsilon_k, \epsilon_\theta(\boldsymbol{a}_t^* + \epsilon_k, k)) \quad (5)$$

where $k$ is the denoising timestep, $\epsilon_k$ is sampled noise, and $\epsilon_\theta(\cdot)$ is the predicted noise from policy. Note that we reuse the RGBD visual tokens computed during rollout rather than recomputing them during the update stage, which substantially speeds up training and enables larger batch sizes. For the critic network, we supervise $V_\theta$ with an MSE loss using the observations $o$, expert trajectories $a$, and the target safety values $v'$:

$$\mathcal{L}^{\text{critic}} = \text{MSE}(v', V_\theta(o, a)). \quad (6)$$

The overall training loss is formulated as:

$$\mathcal{L} = \mathcal{L}^{\text{actor}} + \lambda \mathcal{L}^{\text{critic}}, \quad (7)$$

where $\lambda$ denotes the weight of the critic loss.

### 3.3. Scene preparation and initialization

We build our training environments on the 3D-Front dataset (Fu et al., 2021), which provides large-scale indoor 3D assets. Raw meshes are processed to ensure visual diversity, navigational correctness, and efficient large-scale training in IsaacSim. This section describes our scene processing pipeline, domain randomization strategy, and evaluation benchmark.

**Scene setup.** Our scene pipeline addresses visual diversity, geometric correctness, training efficiency, and rendering fidelity. We use BlenderProc (Denninger et al., 2023) to programmatically re-texture 3D-Front objects without altering geometry. To ensure navigability, we correct outward-facing normals in raw meshes. For efficient large-scale training, we normalize scene heights, vertically stack multiple rooms into

compound assets, and convert them to USD format for IsaacLab. Within the simulator, we enable global illumination, reflections, shadows, ambient occlusion, and advanced antialiasing (DLAA) with a deep learning denoiser to achieve high-quality rendering efficiently.

**Domain randomization.** To improve robustness and generalization, we apply domain randomization at the start of each episode. We sample start–goal pairs using the expert planner and randomize the agent's initial orientation to encourage goal-directed behavior. We additionally vary the agent's height and radius, which affect planning and collision checking, and adjust the camera pose accordingly to simulate diverse viewpoints.

**Evaluation benchmark.** We introduce a benchmark based on 3D-Front for fair, consistent, and thorough evaluation of navigation policies across scene complexity and task difficulty. We curate 8 scene assets that are (i) large enough for long-horizon navigation and (ii) sufficiently cluttered to pose meaningful obstacle-avoidance challenges. All scenes are processed with our high-fidelity pipeline, ensuring visual realism and geometric fidelity. For each scene, we sample 100 start–goal pairs spanning from easy to hard, with difficulty governed by the number of keypoints; these pairs are fixed and consistently used during evaluation to ensure fair comparison across models. Importantly, all evaluation scenes are entirely held out during training and are never seen by the agent.

## 4. Experiments

### 4.1. Implementation details

Our model predicts 24-step waypoint sequence and is initialized with the pretrained weights from NavDP (Cai et al., 2025). As for our expert planner, we perform a short line search in the direction opposite the closest obstacle within 0.1 m, and uniformly resample the raw path at 0.25 m spacing. For each episode, we keep start–goal pairs whose raw path contains at least 5 keypoints to ensure sufficient difficulty. For online data aggregation, each iteration executes 128 rollout steps across 256 parallel environments. Actions are taken from the policy with probability $\rho = 0.8$, otherwise the expert trajectory is executed. For the target safety score, we set $d_{\text{safe}} = 0.5$ and $\alpha = 0.1$. In the subsequent update stage, we train on the newly aggregated data for 10 epochs with a batch size of 2048 and a learning rate of 1e-5. We use Dingo as our agent in IsaacLab. The onboard camera height is randomly sampled in $(0.25, 1.25)$, and the pitch angle is randomly sampled in $(-30°, 0°)$. The radius of the agent is set to 0.25 m with a height equal to the camera. We use 50 scenes to train our model for 1,000 iterations on 8 RTX 4090 GPU, which takes 2 days to complete.

### 4.2. Evaluation and metrics

For the simulation benchmarks, we evaluate our method against four baselines on the NavDP benchmark (Cai et al., 2025) as well as on our curated benchmark for zero-shot evaluation: DD-PPO (Wijmans et al., 2019), iPlanner (Yang et al., 2023), ViPlanner (Roth et al., 2024), and NavDP (Cai et al., 2025). All baselines are used off the shelf, without fine-tuning, to ensure a fair zero-shot comparison. We report two widely adopted metrics for point-goal navigation: Success Rate (SR) and Success-weighted Path Length (SPL). SR measures the fraction of episodes in which the agent stops within 1 m of the goal; an episode is considered a failure if the robot does not reach the goal or becomes stuck due to a collision. SPL evaluates navigation efficiency by weighting successful episodes according to the ratio between the shortest-path distance and the actual path length, averaged over all episodes. For real-world experiments, we compare our method with NavDP using Success Rate.

### 4.3. Results analysis

**Results on NavDP benchmark.** We first evaluate on the NavDP benchmark (Cai et al., 2025). Compared with other baselines, our approach achieves improvements across all metrics and most scenes. In particular, by analyzing the rollouts of NavDP, we observe that its generated trajectories exhibit high variance, resulting in noticeable shaking behavior, as shown in Figure 6. This instability becomes critical near obstacles, often causing failure episodes, which may be attributed to the scarcity of such challenging cases in its training data. In contrast, trajectories produced by *NavOL* remain stable even when moving very close to obstacles, enabling safe and smooth execution. Moreover, on the NavDP benchmark, the performance gains of our method are somewhat less pronounced due to several unfavorable initial configurations, where agents frequently start from positions that immediately lead to falls or collisions. To overcome these, we additionally introduce our benchmark, which encompasses diverse environments with rich layouts and challenging trajectories for a comprehensive evaluation.

**Results on NavOL benchmark.** To comprehensively evaluate the robustness of the navigation policy, we introduce a new benchmark based on the 3D-Front dataset (Fu et al., 2021), selecting scenes with rich spatial layouts and photorealistic textures. The benchmark consists of 8 out-of-domain scenes for zero-shot evaluation. We observe that DD-PPO (Wijmans et al., 2019) consistently fails to find feasible paths to the target in these complex scenes, while both iPlanner (Yang et al., 2023) and ViPlanner (Roth et al., 2024) frequently get stuck around corners or struggle to circumvent structural obstacles such as walls. NavDP (Cai et al., 2025) also shows limited performance, mainly due to inconsistent data distribution. As shown in Table 2, our method

*Table 1.* **Performances on the NavDP benchmark (Cai et al., 2025).** The results show that our approach consistently outperforms all baselines across scenes and metrics, particularly in SPL, indicating a stronger ability to approximate optimal trajectories.

| Methods | Cluttered-Easy | | Cluttered-Hard | | Intern-Home | | Intern-Commercial | | Average | |
| --- | --- | --- | --- | --- | --- | --- | --- | --- | --- | --- |
| | SR($\uparrow$) | SPL($\uparrow$) | SR($\uparrow$) | SPL($\uparrow$) | SR($\uparrow$) | SPL($\uparrow$) | SR($\uparrow$) | SPL($\uparrow$) | mSR($\uparrow$) | mSPL($\uparrow$) |
| DD-PPO | 0.0 | 0.0 | 0.0 | 0.0 | 0.4 | 0.4 | 5.3 | 5.2 | 1.4 | 1.4 |
| iPlanner | 89.4 | 88.4 | 80.3 | 78.8 | 43.0 | 40.6 | 54.6 | 52.8 | 66.8 | 65.1 |
| ViPlanner | 80.2 | 80.1 | 64.7 | 64.5 | 45.0 | 43.2 | 63.7 | 61.9 | 63.4 | 62.4 |
| NavDP | 92.3 | 90.5 | 87.4 | 84.9 | **60.0** | **55.6** | 71.4 | 68.2 | 77.8 | 74.8 |
| Ours | **95.6** | **92.0** | **92.0** | **87.3** | 58.8 | 54.9 | **75.2** | **71.1** | **80.4** | **76.3** |

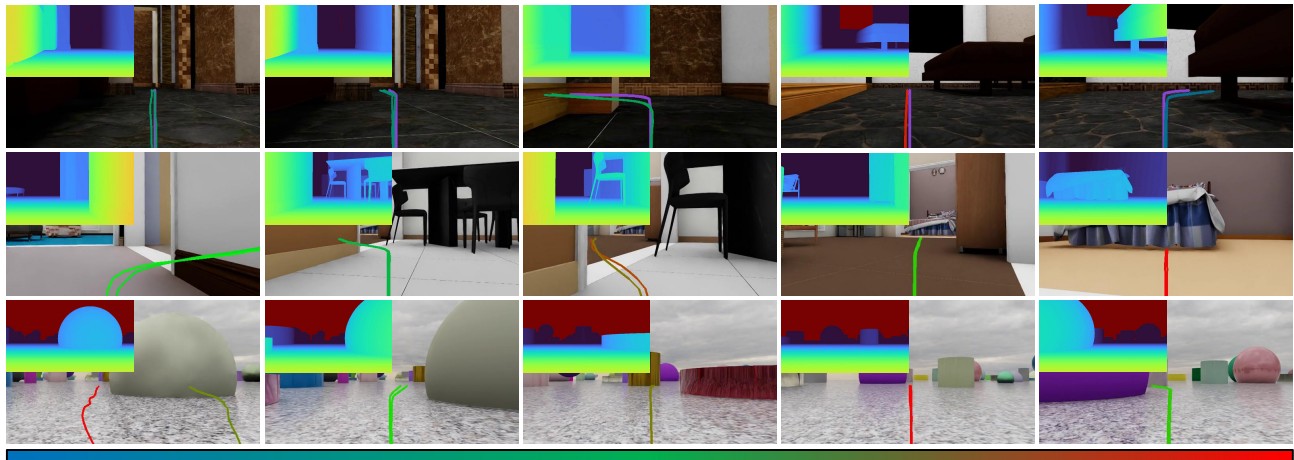

Dangerous (low critic value)        Safe (high critic value)

*Figure 5.* Visualization of predicted trajectories projected into the camera frame. Blue line denotes dangerous trajectories with low critic values while red line denotes safe trajectories with high critic values. The first row shows visualizations in a training scene, where ground-truth meshes let us render expert trajectories (purple). The second and third rows show results on the benchmark.

*Table 2.* **Performances on our benchmark.** Our model substantially outperforms all baselines across almost all scenes while exhibiting stronger stability, highlighting the robustness and generalization of our on-policy online imitation learning framework in minigating distributional shift.

| Methods | Scene 1 | | Scene 2 | | Scene 3 | | Scene 4 | | Scene 5 | | Scene 6 | | Scene 7 | | Scene 8 | | Avg | |
| --- | --- | --- | --- | --- | --- | --- | --- | --- | --- | --- | --- | --- | --- | --- | --- | --- | --- | --- |
| | SR($\uparrow$) | SPL($\uparrow$) | SR($\uparrow$) | SPL($\uparrow$) | SR($\uparrow$) | SPL($\uparrow$) | SR($\uparrow$) | SPL($\uparrow$) | SR($\uparrow$) | SPL($\uparrow$) | SR($\uparrow$) | SPL($\uparrow$) | SR($\uparrow$) | SPL($\uparrow$) | SR($\uparrow$) | SPL($\uparrow$) | mSR($\uparrow$) | mSPL($\uparrow$) |
| DD-PPO | 1.0 | 0.9 | 1.0 | 1.0 | 9.0 | 8.2 | 0.0 | 0.0 | 1.0 | 0.7 | 4.0 | 3.0 | 3.0 | 2.0 | 3.0 | 2.7 | 2.9 | 2.4 |
| iPlanner | 3.0 | 2.9 | 25.0 | 24.6 | 10.0 | 9.6 | 27.0 | 26.9 | 24.0 | 23.9 | 27.0 | 26.4 | 26.0 | 25.5 | 6.0 | 5.9 | 18.7 | 18.4 |
| ViPlanner | 14.0 | 12.7 | 31.0 | 29.9 | 23.0 | 22.0 | 42.0 | 41.0 | 43.0 | 41.5 | 50.0 | 46.9 | 37.0 | 35.1 | 22.0 | 20.9 | 33.0 | 31.5 |
| NavDP | 41.0 | 33.9 | 36.0 | 31.9 | 37.0 | 29.9 | 37.0 | 33.3 | 25.0 | 23.0 | 25.0 | 22.3 | 50.0 | 44.3 | **87.0** | **83.3** | 42.2 | 37.7 |
| Ours | **54.0** | **43.7** | **69.0** | **62.9** | **80.0** | **76.6** | **75.0** | **72.6** | **72.0** | **70.0** | **77.0** | **72.2** | **70.0** | **64.5** | 55.0 | 47.0 | **69.0** | **63.7** |

achieves state-of-the-art performance on most scenes, surpassing the strongest baseline, NavDP (Cai et al., 2025), validating the effectiveness of our interactive online training paradigm and the robustness and generalization capability of our online imitation learning framework in mitigating distributional shift. Furthermore, Figure 5 visualizes the trajectories predicted by *NavOL*, demonstrating its ability to generate robust trajectories even in zero-shot scenarios.

**Real-world results** We compare the performance of NavDP and our method on a Unitree Go2 robot equipped with a RealSense D435i camera for zero-shot sim-to-real evaluation. As shown in Figure 7a, we evaluate both meth-

ods in three scenarios, Office, Gym, and Corridor, designed to assess obstacle avoidance under cluttered environments. For each scene, we conduct 10 trials per method, and report the overall success rate in Table 3. Our method consistently outperforms NavDP across all scenarios. While NavDP frequently becomes stuck in complex environments, our approach is able to identify safe trajectories and successfully navigate around obstacles, even when operating in close proximity to them. Notably, the robot used during simulation training is Dingo, which differs from the Unitree Go2 platform used for real-world deployment. This result demonstrates that our policy exhibits strong cross-embodiment generalization, having learned transferable spatial represen-

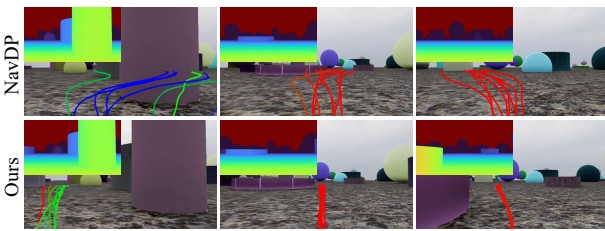

*Figure 6.* Qualitative comparison of sampled trajectories on three matched start–goal pairs. The trajectory distributions of NavDP (top) disperse in the vicinity of obstacles, with multiple samples crossing them, whereas those of *NavOL* (bottom) remain concentrated along a collision-free path.

*Table 3.* Real-world evaluation results on Unitree Go2, reported in terms of success rate (successful trials / total trials).

| Methods | Office | Gym | Corridor |
|---|---|---|---|
| NavDP | 4/10 | 2/10 | 2/10 |
| Ours | **8/10** | **7/10** | **10/10** |

*Table 4.* Ablation studies on different components of *NavOL*.

| Components | SR($\uparrow$) | SPL($\uparrow$) |
|---|---|---|
| Train with 1 scene | 57.8 | 53.4 |
| Train with 10 scene | 64.8 | 60.5 |
| Rollout 8 steps | 67.9 | 63.1 |
| Rollout 32 steps | 68.5 | 61.6 |
| w/o expert rollout | 43.9 | 41.4 |
| From scratch | 48.3 | 45.5 |
| w/o path processing | 64.2 | 59.2 |
| w/o camera rand | 66.0 | 59.2 |
| w/o critic | 67.1 | 61.6 |
| Full | **69.0** | **63.7** |

tations and robust navigation behaviors that enable effective adaptation to previously unseen environments. Figure 7b additionally shows the onboard RGB sequence captured by the Go2 along these rollouts.

## 4.4. Ablation study

In this section, we ablate the main components of our approach, number of training scenes, rollout configuration, training from scratch, domain randomization, and the critic network. All results are reported using our benchmark.

**Number of training scenes.** We vary the number of 3D scenes used to train *NavOL*. The 1-scene setting uses a single scene, while the 10-scene setting trains on 10 distinct scenes. As shown in Table 4, both SR and SPL consistently improve as the number of training scenes increases, indicating strong benefits from scaling scene diversity. Notably, the "1 scene" model achieves a 90% success rate on its training scene and still outperforms NavDP (Cai et al., 2025), suggesting it

*Table 5.* Compute and performance comparison with NavDP on our benchmark. NavDP GPU-day estimates are taken from its paper.

| Method | mSR | mSPL |
|---|---|---|
| NavDP (original, $32\times$A100, 32 days) | 42.2 | 37.7 |
| *NavOL* from scratch ($8\times$4090, 2 days) | 48.3 | 45.5 |
| *NavOL* (NavDP init, $+8\times$4090, 2 days) | **69.0** | **63.7** |

does not simply overfit on training scene.

**Rollout.** We study the effect of rollout length and a "w/o expert rollout" variant ($\rho = 1$), executing only policy actions. Table 4 shows similar performance across different rollout lengths, likely because episodes average 400 steps, so varying per-iteration rollout steps yields comparable data per episode. In contrast, removing expert executions degrades performance, as purely policy rollouts often visit undesirable states, making optimization harder.

**Train from scratch.** We also try to initialize *NavOL* from scratch without using the pretrained weights from NavDP (Cai et al., 2025), performance drops substantially versus the full model. The outcome is unsurprising, without a good prior, early rollouts are unstable, which hampers supervision quality and leads to difficult optimization.

**Domain randomization.** We further ablate two components on domain randomization. "w/o path processing" removes the local line search used by the expert planner described in Section 3.2. Without this step, the policy tends to skim obstacles and is more likely to get stuck near corners. "w/o camera rand" fixes the height and pitch of the camera. As shown in Table 4, enabling camera randomization improves zero-shot generalization across scenes.

**Critic network.** Finally, we assess test-time trajectory selection with the critic. In our online imitation setting, the critic yields only modest gains, likely because the policy already learns to avoid unsafe trajectories during on-policy training. Nonetheless, the critic provides a lightweight safety margin and occasionally corrects risky choices.

## 5. Discussion

**Map dependence and compute fairness.** At inference, *NavOL* deploys zero-shot with no map required; NavMesh is used *only* during training to supply expert path labels, and 3D asset preparation is a one-time preprocessing step amortized across runs. To assess whether comparing *NavOL* (initialized from NavDP) against NavDP is compute-fair, we separate the two regimes in Table 5. NavDP requires roughly 1024 GPU-days; the from-scratch variant of *NavOL* uses only 16 GPU-days yet already attains mSR=48.3 versus NavDP's 42.2, indicating that on-policy online supervision is far more data-efficient than offline rendering of massive

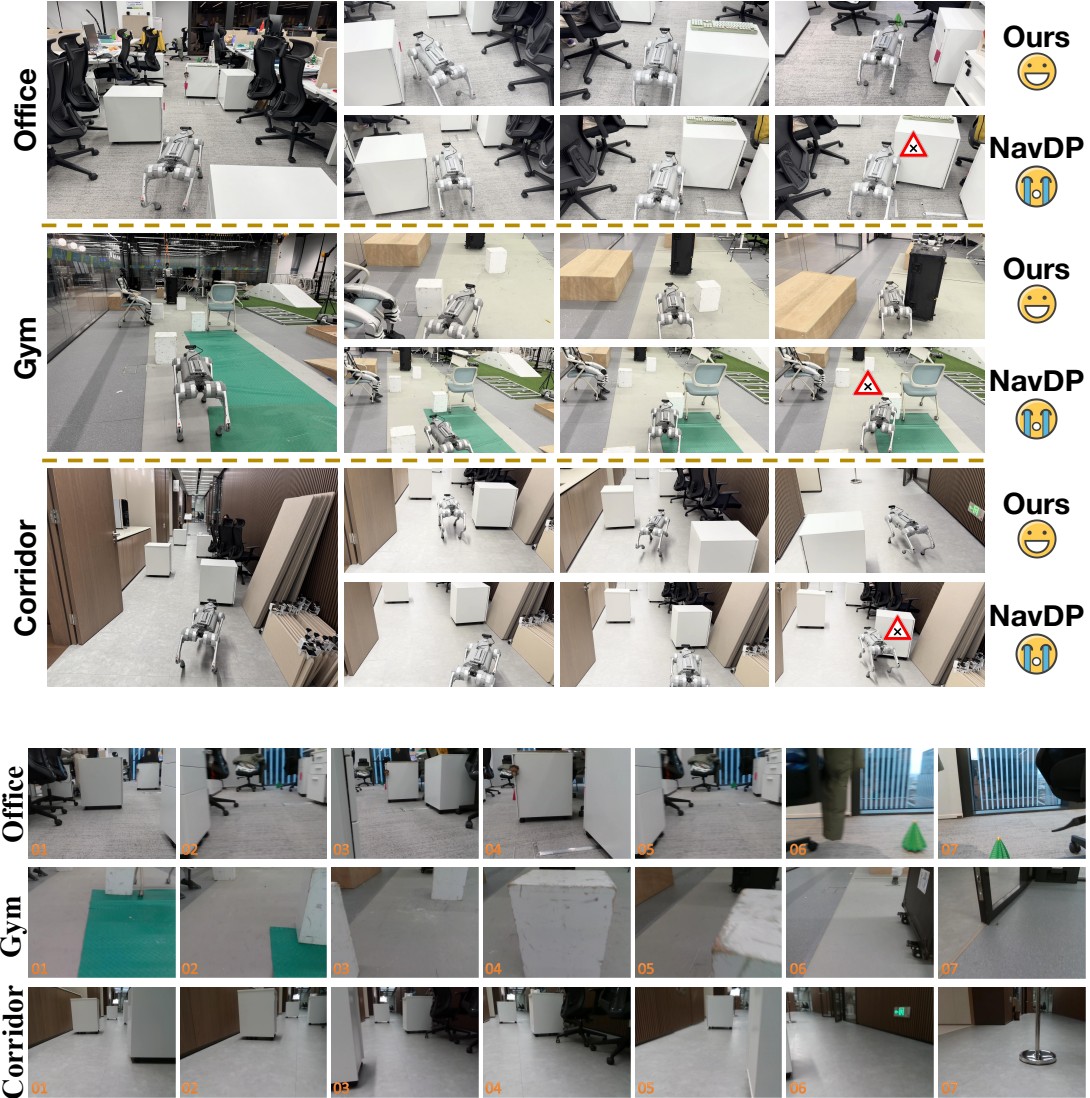

*Figure 7.* Zero-shot real-world deployment on a Unitree Go2 across three cluttered scenes – *Office*, *Gym*, and *Corridor*. *Top* (a): third-person rollouts – *NavOL* (top sub-row of each scene) reaches the goal, NavDP (bottom sub-row) fails. *Bottom* (b): onboard RGB snapshots (01–07) captured along the same rollouts.

trajectory datasets. With NavDP initialization plus another 16 GPU-days, mSR rises to 69.0 (+26.8 at ∼1.5% extra compute), indicating that the online phase is the source of the improvement rather than the initialization.

## 6. Conclusion

We present *NavOL*, an online imitation learning framework for robust visual navigation that bridges the gap between offline imitation learning and online reinforcement learning. Unlike conventional imitation methods that rely on static, pre-collected datasets, *NavOL* continuously collects expert trajectories from a privileged global planner and updates a navigation policy in a rollout–update loop. This design eliminates the need for manual reward engineering, improves data efficiency, and mitigates distribution shift by training on the policy's own rollouts. Leveraging IsaacLab for fast, high-fidelity parallel simulation, *NavOL* scales to large, diverse scene collections and efficiently acquires long-horizon trajectories. We also propose an indoor navigation benchmark that provides a comprehensive testbed for assessing robustness and generalization, on which *NavOL* consistently achieves strong performance across scenarios. Real-world evaluations further demonstrate the effectiveness of the proposed online imitation learning pipeline.

## Acknowledgements

This work was supported in part by New Generation Artificial Intelligence-National Science and Technology Major Project (2025ZD0123004), Ningbo grant (2025Z038) and National Natural Science Foundation of China (Grant No. 62376060).

## Impact Statement

This paper focuses on advancing online imitation learning for visual navigation. By proposing a pipeline that mitigates the distribution shift of offline methods and avoids the reward engineering of reinforcement learning, we aim to lower the barriers to training robust embodied agents. Our work primarily impacts the research community by providing a more data-efficient and scalable alternative to current policy learning paradigms. We do not foresee negative societal consequences that must be specifically highlighted here.

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

## A. More results

**Benchmark**    For each scene in the benchmark, both in-domain and out-of-domain, we randomly sample start and goal positions as described in Section 3.2. Using the expert planner (Figure 8), we then generate a globally optimal path to ensure that navigation from start to goal is feasible.

**Visualization of projected trajectories**    We further visualize the predicted trajectories projected into the camera frame, as shown in Figure 9, Figure 10, Figure 11, and Figure 12. These qualitative results show that our policy produces stable, collision-avoiding paths, even in cluttered layouts and near obstacles. The trajectories remain smooth and directed toward the goal, supporting our quantitative findings on reliability and generalization across both NavDP benchmark and our own benchmark. Accompanying videos are included in the supplementary material.

**Real-world deployment on Go2**    To validate the effectiveness of our approach in physical environments, we deployed the trained policy on a Unitree Go2 quadruped robot. Figure 13 visualizes the camera views captured by the onboard RealSense camera during these deployment trials. The figure displays sequences from three challenging environments: *Office*, *Gym*, and *Corridor*. As observed in the snapshots, the Go2 robot successfully navigates through various environments while effectively avoiding obstacles, demonstrating robust obstacle avoidance across diverse scene styles. These real-world trials validate that our system can robustly process diverse scenes in the real world, supporting the generalization capabilities derived from massively parallel training in Isaac Lab and data-intensive online imitation learning. Accompanying videos are included in the supplementary material.

## B. Limitations and Future Work

**Asset pipeline.**    Our scene preparation pipeline, based on BlenderProc and IsaacSim, reduces accessibility, which we identify as a limitation of the present work. This overhead, however, is confined to the one-time preparation of training data and does not affect deployment, since at inference time the trained policy is applied zero-shot to new scenes and requires no map (Section 5).

**Training compute cost.**    Training *NavOL* on $8\times$RTX 4090 GPUs for 2 days remains a meaningful hardware requirement, marking a further practical limitation of our method. Although our compute budget is substantially smaller than that of NavDP, it remains non-negligible in absolute terms.

**Real-world failure modes and future directions.**    We further analyze failure modes observed during real-world deployment on the Unitree Go2. *(i) Scene-edge exploitation:* in cluttered scenes such as the Office setting, the policy occasionally identifies boundary gaps of the test area as passable paths, driving the robot along edges or into peripheral gaps and preventing it from reaching the goal. *(ii) Low-obstacle misjudgment:* owing to the limited mounting height of the onboard camera, the robot sometimes fails to perceive the full geometry of low obstacles such as table legs or stool legs, attempting to pass underneath and becoming stuck. These failure modes point to clear directions for future work: multi-height obstacle perception, and scene-boundary robustness.

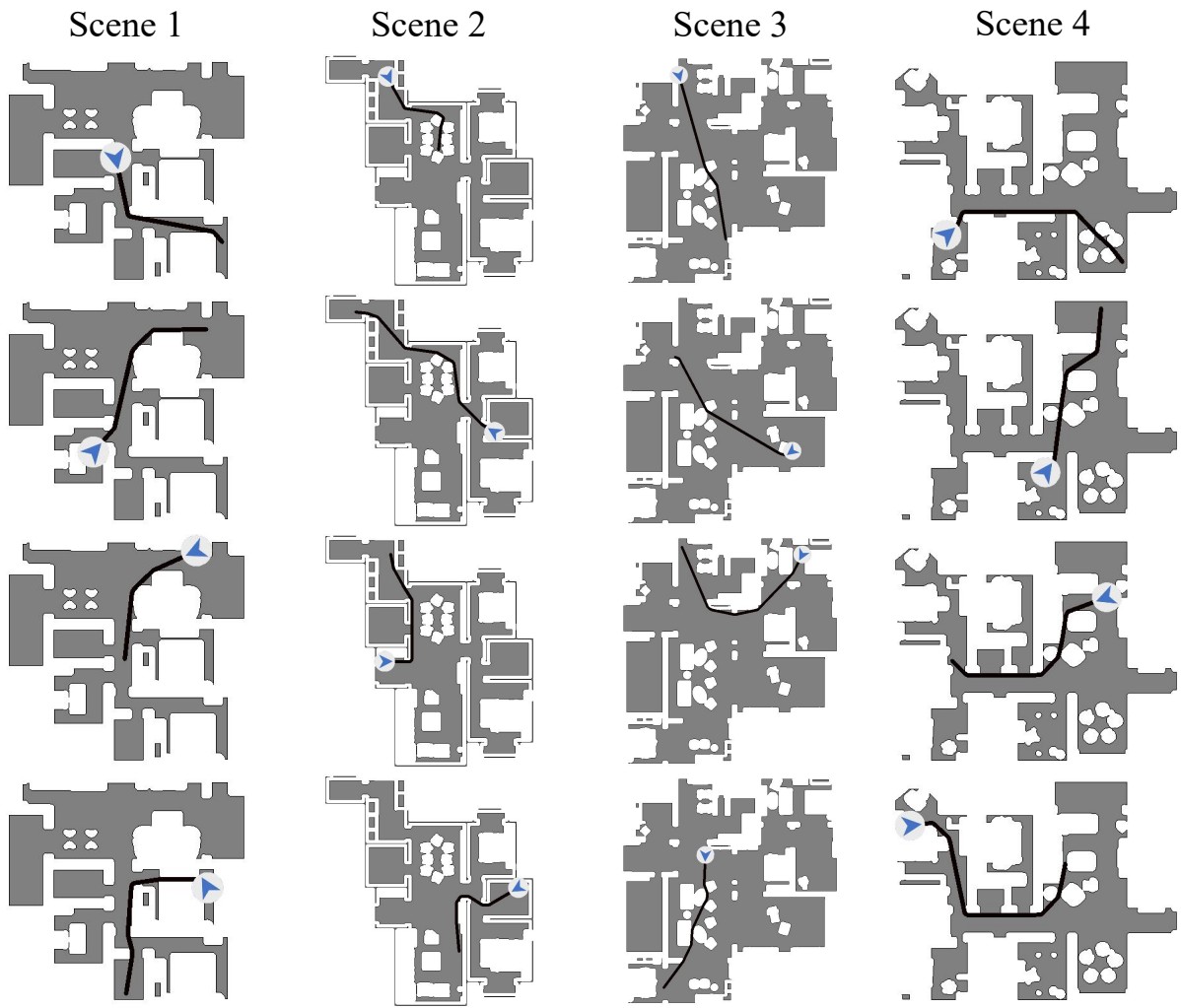

*Figure 8.* Trajectories planned by the expert planner in our benchmark.

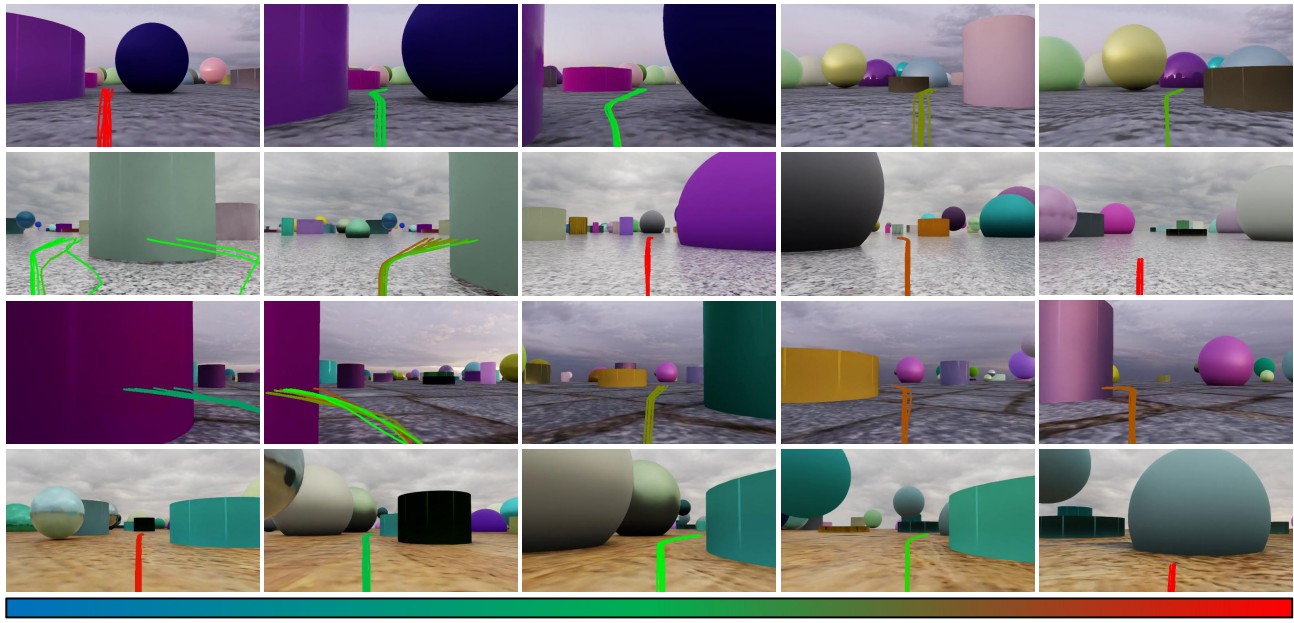

Dangerous (low critic value)         Safe (high critic value)

*Figure 9.* Visualization of predicted trajectories projected into the camera frame.

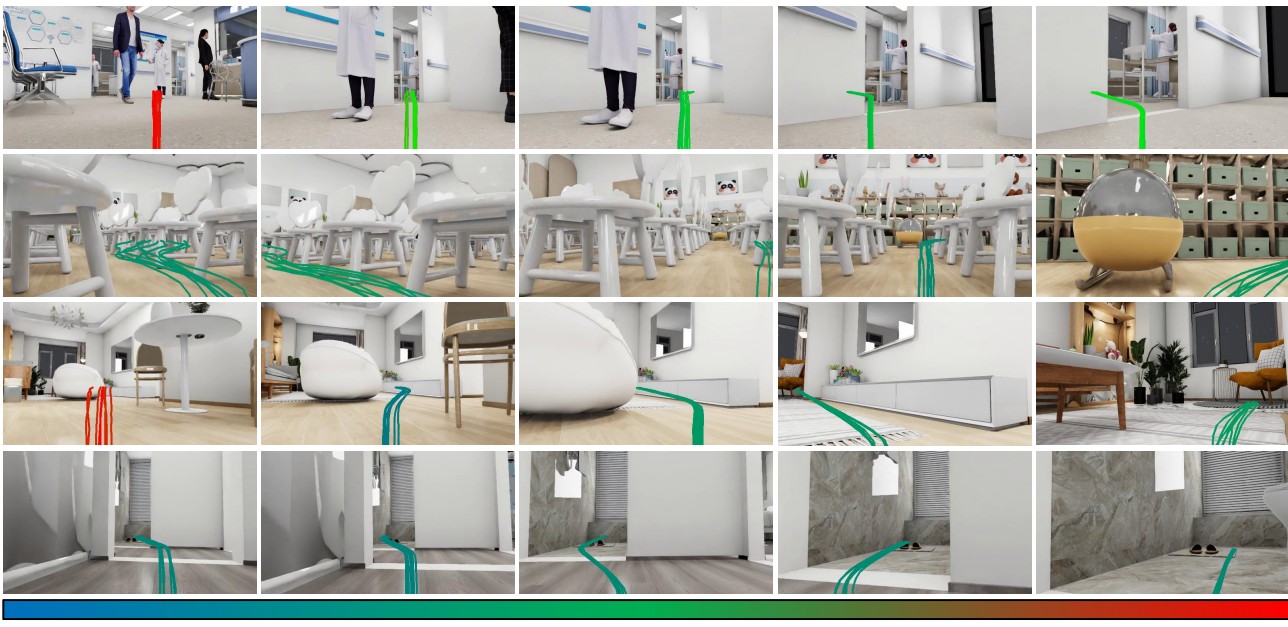

Dangerous (low critic value)         Safe (high critic value)

*Figure 10.* Visualization of predicted trajectories projected into the camera frame.

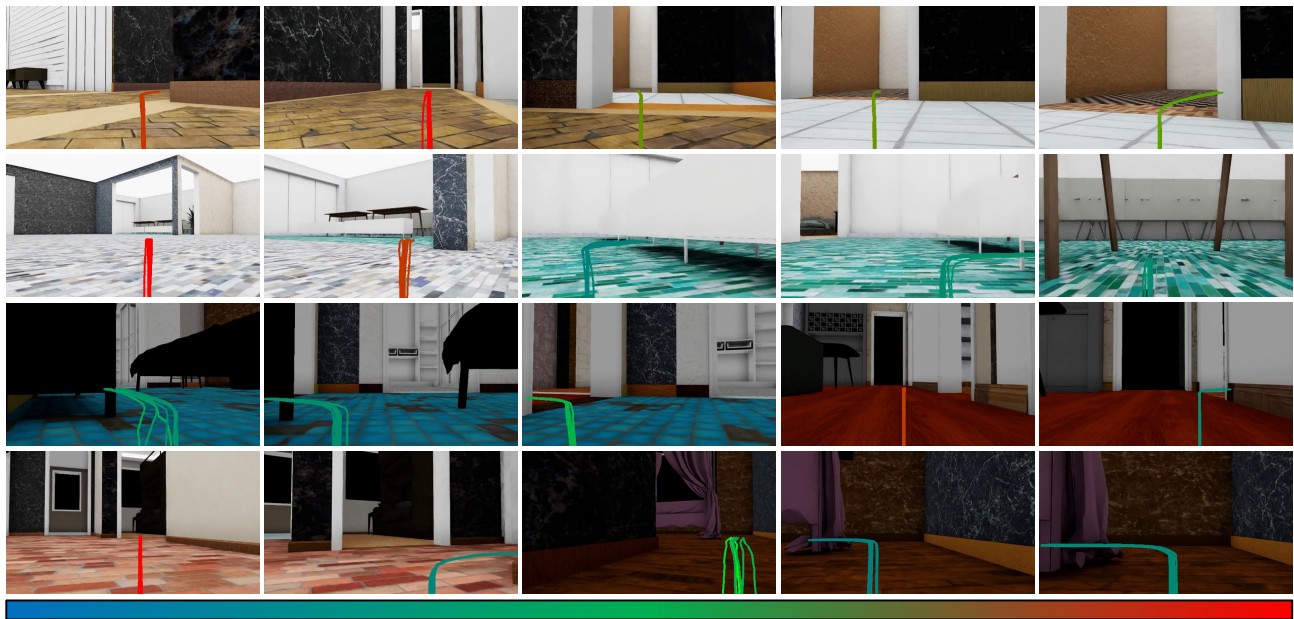

Dangerous (low critic value)                                    Safe (high critic value)

*Figure 11.* Visualization of predicted trajectories projected into the camera frame.

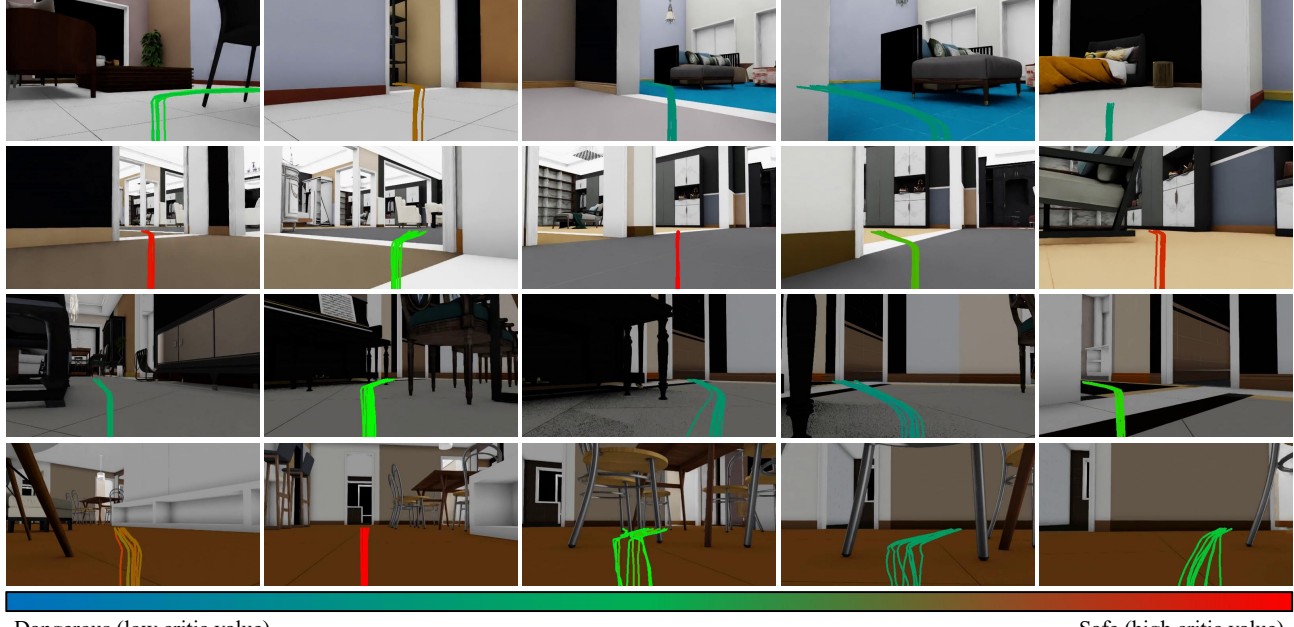

Dangerous (low critic value)                                    Safe (high critic value)

*Figure 12.* Visualization of predicted trajectories projected into the camera frame.

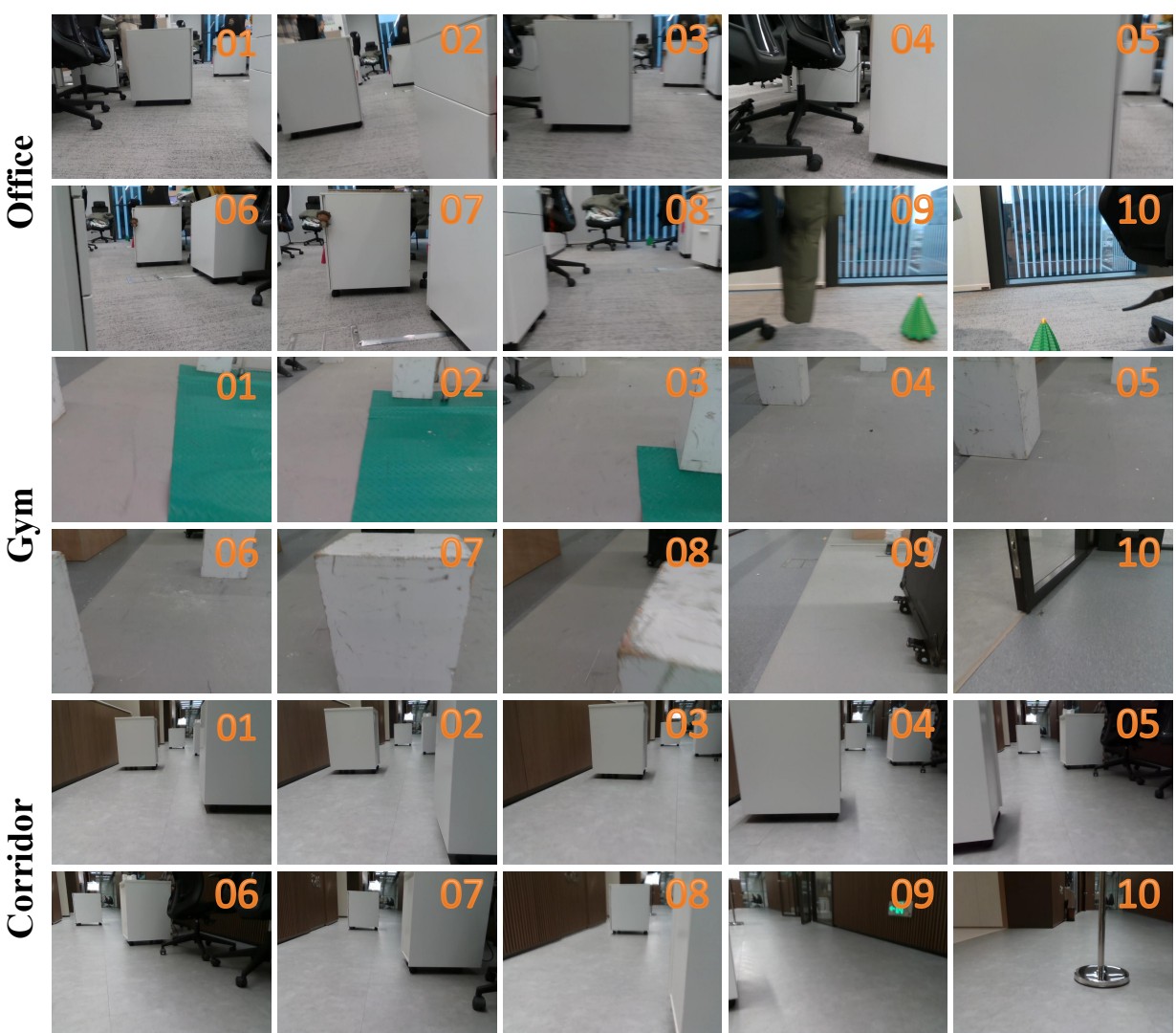

*Figure 13.* Visualization of the view from the Go2 RealSense camera during real-world deployment.

