# OpenReview forum: "NavOL: Navigation Policy with Online Imitation Learning"
_ICML.cc/2026/Conference — ICML 2026 regular_

### Official Review · Reviewer_FLCJ · 2026-02-23

**Soundness:** 3
**Presentation:** 3
**Significance:** 3
**Originality:** 3
**Overall Recommendation:** 5
**Confidence:** 4

**Summary:**

In this work authors emphasize the importance of navigation as an important robotics challenge. To improve over existing offline Imitation Learning methods, authors propose an active learning system (A DAgger-like method adapted for large visual diffusion policies). Authors focus on the problem of goal-conditioned trajectory planning, and assume access to a low level controller (e.g. MPC). Assuming access to expert waypoint trajectories, authors are able to showcase that their iterative learning approach NavOL outperforms existing offline baselines regarding zero-shot performances on unseen scenes. This evaluation is conducted both on held out simulated scenes and on real world scenarios.

**Compliance With Llm Reviewing Policy:**

Affirmed.

**Final Justification:**

Other reviews did not showcase hidden limitations, the rebuttal was convincing and reinforced my decision: I will maintain my accept.

**Key Questions For Authors:**

Question are highlighted in the Strengths and Weaknesses paragraph

**Limitations:**

I would have prefered to see a bit more discussion on the failure cases of the method, especially in the real world experiments.

**Strengths And Weaknesses:**

## Soundness
While experiments seem quite sound (adequate experiments, ablations, baselines), I have a few concerns. **How many training repeats (aka seeds) are being used to compare NavOL and baselines ?**

Also, minor but: it would have been interesting to see or discuss failure cases in the real world scenarios. **From the videos in the supplementary material, I see that the real-world quadrupedal robot is successful at navigating relatively straight paths including static obstacle avoidance. What about 90 or 180 degrees turns ?**


## Presentation
The paper is well structured, well written. I enjoyed reading it.


## Significance
pros: authors conducted zero-shot evaluations on both held-out simulated scenes and on a real world robot, which is a convincing demonstration of their approach.

cons: Authors do not share the code in the supplementary nor mention they will open source it in the manuscript. Active learning frameworks can be quite complex to setup, and as such it is highly valuable to share the code, when possible. Especially when claimed contributions include the implementation of new parallelized active learning pipeline and a new visual indoor navigation benchmark. **If this work was accepted, would you release your codebase, including your optimized active learning pipeline, model checkpoints and training/eval code ?**

## Originality
I do not have a clear picture of the state of the art regarding DiT + Active Learning + Robotics, so it is hard for me to say if the work is novel or not.

However, I wonder how different their approach is from the original DAgger algorithm.**For me, if we abstract from the policy (which here is a DiT with a safety mechanisms using a scoring head), it is exactly the DAgger method. Is this understanding correct ?**

I also wonder how novel is the use of a safety scoring head in the safe RL litterature.

Even if the methods are not novel however, this paper still presents an interesting work, combining efficiently active learning methods and safety methods into a navigation planner.

---

> ### Author Rebuttal · Authors · 2026-03-31
>
> We thank the reviewer for the careful reading and are glad that the paper's structure provided a clear reading experience. We respond to each point below.
>
> **Soundness — Training seeds.**
>
> We further ran the full model with **3 independent seeds** and report mean $\pm$ std: mSR = 69.2 $\pm$ 0.7, mSPL = 63.5 $\pm$ 0.8. Given our $\sim$27-point mSR advantage over the strongest baseline NavDP, the ranking is highly stable across seeds.
>
> **Soundness — 90° and 180° turns.**
>
> Our model can indeed perform 90° and 180° turns (e.g., turning through corridor corners toward an elevator, or starting with the heading facing away from the goal), though the rebuttal format does not directly support image uploads. This capability stems from training requiring $F \geq 5$ heading changes per episode, many involving multi-angle turns as shown in Figure 5. Benchmark episodes traverse multiple rooms and corridor intersections, inherently involving numerous turns, which is a key reason we outperform baselines.
>
> **Soundness — Failure cases in real-world.**
>
> We appreciate this suggestion. Typical failure modes observed during deployment include:
>
> (1) **Scene-edge exploitation**: in cluttered scenes (e.g., Office), the policy occasionally identifies boundary gaps of the test area as passable paths, driving the robot along edges or into peripheral gaps, preventing goal arrival.
>
> (2) **Low-obstacle misjudgment**: limited by the camera mounting height, the robot sometimes cannot perceive the full geometry of low obstacles (table legs, stool legs), attempting to pass underneath and getting the chassis stuck.
>
> These failures point to clear future directions: multi-height obstacle perception and scene-boundary robustness.
>
> **Significance — Code release.**
>
> **Yes, we commit to releasing the full codebase upon acceptance**, including the IsaacLab-based online IL pipeline, NavMesh expert planner, NavOL checkpoints, evaluation code, and scene scripts (BlenderProc + USD conversion). Our system spans scene processing, simulation training, and real-robot deployment, a substantial end-to-end effort.
>
> **Originality — Is NavOL just DAgger?**
>
> We are indeed inspired by DAgger, but there is a fundamental difference: prior DAgger-style algorithms have been limited to simple controllers, and scaling to visual diffusion policies has remained computationally prohibitive. Our core contribution is **demonstrating that DAgger can scale to visual diffusion policies**, through the following designs that resolve the scalability bottleneck:
>
> 1. **Visual token reuse**: ViT features computed once during rollout are cached across all DDPM denoising steps and update epochs, reducing much compute.
> 2. **Current-rollout-only data**: fixed compute per iteration, avoiding the dataset bloat of offline methods.
> 3. **Shared-backbone critic**: actor and critic share the ViT/DiT backbone, no extra forward pass at inference.
>
> Together these enable 256 parallel environments $\times$ 50 scenes at 2000+ trajectories/hour on 8 RTX 4090 GPUs, unprecedented for DAgger-based robotics.
>
> **Originality — Novelty of the safety scoring head.**
>
> Our critic differs from Safe RL approaches in two key aspects: (1) it is supervised by **geometric prior labels** (NavMesh obstacle distances) rather than learned from reward signals, requiring only two fixed hyperparameters ($d_{\text{safe}}$, $\alpha$) with no environment-specific tuning; (2) it serves as a **test-time trajectory ranker** that selects the safest among diffusion-sampled candidates, rather than constraining policy optimization.The critic shares the ViT/DiT backbone with the actor and is trained jointly via $\mathcal{L} = \mathcal{L}^{\text{actor}} + \lambda\, \mathcal{L}^{\text{critic}}$. Table 2 confirms that removing it ("w/o critic") only decreases SR from 69.0 to 67.1 (1.9-point drop) with no impact on training stability, showing the critic is an **auxiliary enhancement**; the policy's core capability comes from $\mathcal{L}^{\text{actor}}$ **(online expert supervision)**.

---

> > ### Author Rebuttal · Reviewer_FLCJ · 2026-04-01
> >
> > I thank the authors for their answers, which are reassuring, especially the release of the codebase. No more questions. I will increase my score to accept. This work is ICML-worthy.

---

> > > ### Author Response · Authors · 2026-04-04
> > >
> > > We sincerely thank reviewer for the encouraging assessment and for considering our work ICML-worthy. We reaffirm our commitment to releasing the full codebase upon acceptance, including the IsaacLab-based online IL pipeline, NavMesh expert planner, model checkpoints, evaluation code, and scene processing scripts.

---

### Official Review · Reviewer_LD49 · 2026-03-04

**Soundness:** 3
**Presentation:** 3
**Significance:** 3
**Originality:** 3
**Overall Recommendation:** 4
**Confidence:** 4

**Summary:**

The paper presents NavOL, an online imitation learning for training robust and general visual navigation policies. The core contribution is a rollout–update loop where a pre-trained navigation diffusion policy interacts with a high-fidelity simulator (IsaacLab) and receives real-time supervision from a privileged global planner.

By online imitation learning, NavOL mitigates the distribution shift and compounding errors typical of offline imitation learning, while avoiding the complex reward engineering associated with reinforcement learning. The system is highly scalable, capable of collecting over 2,000 high-quality trajectories per hour using parallel GPU simulation. The authors also introduce a new indoor navigation benchmark and demonstrate successful zero-shot Sim2Real transfer to a real quadruped robot.

**Compliance With Llm Reviewing Policy:**

Affirmed.

**Final Justification:**

My main concerns have been addressed in the rebuttal.

**Key Questions For Authors:**

1. **Catastrophic forgetting and data retention**: In Section 3.2, you mention discarding trajectories from previous iterations and retaining only the current rollout data during the update stage to maximize efficiency. Did you observe any issues with catastrophic forgetting of behaviors learned in earlier iterations or different scenes? Why was a replay buffer of past "hard" examples not employed, and how might its absence affect long-term stability?

2. **Expert planner robustness**: The system relies on a privileged NavMesh-based planner. In real-world deployment, NavMesh is unavailable. If the simulator's NavMesh has geometric inaccuracies or if the environment contains unmapped dynamic obstacles (like moving humans), how does the policy handle the discrepancy between the "perfect" simulation supervision and the "imperfect" real world?
This is also a sim-to-real gap.

3. **Inference latency on edge hardware**: The NavOL architecture utilizes a DiT and a ViT backbone (DAV2). What is the actual inference frequency (Hz) on the Unitree Go2’s onboard compute, and how does this affect navigation speed? High-dimensional diffusion models often suffer from high latency which can be detrimental to high-speed obstacle avoidance.

4. **Goal-agnostic critic design**: You state that the safety critic excludes goal tokens to remain goal-agnostic. While this focuses on safety, is there a risk that the critic might rank a "safe" trajectory highly even if it leads the robot significantly away from the target goal? Why was the decision made to separate safety from goal-directedness in the ranking process?

**Limitations:**

Not fully. While the authors include an "Impact Statement", it is relatively brief and primarily claims a lack of foreseeable negative consequences. To meet the high standard of an ICML submission, the authors should be encouraged to add a dedicated limitations section or expand their discussion to include:

- Reliance on pre-training: The method is heavily dependent on the quality of the initial NavDP prior.
- Static indoor environment: The training primarily occurs in static indoor scenes from the 3D-Front dataset. The paper would benefit from discussing how the model might fail or require adaptation in high-traffic dynamic environments (e.g., crowded malls).
- Asset requirements: The method requires a sophisticated pipeline involving BlenderProc for re-texturing and IsaacSim for high-fidelity rendering. Discussing the accessibility of these resources for the broader research community would provide a more balanced view of the work's impact.
- Hardware constraints: The computational cost of running the rollout-update loop (requiring 8 RTX 4090s for 2 days) is a significant barrier that should be formally acknowledged.

**Strengths And Weaknesses:**

- **Soundness**
  - **Strengths**: The method expands DAgger to visual diffusion policies and shows its effectiveness on sim2real navigation. Experiments are comprehensive and include multiple simulation benchmarks and real-world scenarios. The proposed domain randomization techniques and new navigation benchmark also provide good tools for learning-based navigation.

  - **Weaknesses**:
    1. The underlying method (DAgger) is off-the-shelf and the solution looks straightforward, making this work look more systematic and engineering rather than principled algorithms.
    2. During online imitation, while the "privileged global planner" provides high-quality labels, the paper could more deeply discuss the failure modes if the expert planner itself encounters ambiguous geometry in the NavMesh.
    3. Additionally, the ablation study shows that starting from scratch significantly degrades performance, suggesting the method is heavily dependent on a strong offline-pretrained prior (NavDP).

- **Presentation**:
The paper is overall well-structured and clearly written. The authors provide a thorough related work section that properly positions NavOL between offline IL and RL. Some technical details regarding the "safety score" calculation (Equation 4) should be better augmented with intuitive explanations, like how the hyperparameters affect the score.

- **Significance**
This work addresses the data scarcity bottleneck in robot navigation. By utilizing modern simulators to generate online rollouts and learning from those, it provides a practical path for developing more robust navigation foundation models. This is more like utilizing modern practices for improving a classic domain—navigation—instead of proposing disruptive methods.
The impact might also be somewhat constrained by the requirement for a high-fidelity 3D asset pipeline (like the processed 3D-Front dataset) and significant GPU resources, which may not be accessible to all researchers.

- **Originality**
The algorithm and method overall are not new, DAgger + DiT + IsaacLab + Sim2Real are all existing technologies. The work is more of a system-level innovation and an optimization of the learning paradigm rather than a fundamental change to fundamental algorithms.
Therefore, the primary originality lies in the novel combination of diffusion-based policies with a massively parallel online DAgger loop. The integration of a shared-backbone critic network for real-time trajectory ranking is a clever way to provide an additional safety margin during deployment.

---

> ### Author Rebuttal · Authors · 2026-03-31
>
> We thank the reviewer for recognizing the effectiveness of our method, the well-organized presentation, and the practical value of addressing data scarcity in navigation, and respond below.
>
> **W1 — DAgger is off-the-shelf; the work is more engineering than innovation.**
>
> We are indeed inspired by DAgger, but there is a fundamental difference: prior DAgger-style algorithms have been limited to simple controllers, and scaling to visual diffusion policies has remained prohibitive. Our core contribution is **demonstrating that DAgger can scale to visual diffusion policies**, via:
>
> 1. **Visual token reuse**: ViT features computed once are cached across all DDPM denoising steps and update epochs, reducing compute by $\sim$10$\times$.
> 2. **Current-rollout-only data**: fixed compute per iteration, avoiding dataset bloat.
> 3. **Shared-backbone critic**: actor and critic share ViT/DiT backbone, no extra forward pass at inference.
>
> Together these enable 256 envs $\times$ 50 scenes at 2000+ trajectories/hour on 8 RTX 4090s, unprecedented for DAgger-based robotics.
>
> **W2 / Q2 — Expert planner robustness; NavMesh inaccuracies and dynamic obstacles.**
>
> It is worth noting that NavMesh is used **only during training**; the deployed policy is purely reactive (RGB-D $\to$ waypoints) with no map. We acknowledge NavMesh precision caps expert label quality. To mitigate this, we filtered flawed assets during scene selection and enabled global illumination, reflections, shadows, ambient occlusion, and DLAA to narrow the sim-to-real gap. In real-robot deployment, NavOL shows notably better obstacle avoidance than the offline-only version. For dynamic obstacles, our policy effectively avoids moving pedestrians in the real world, as the RGB-D input naturally captures visible moving objects that influence trajectory predictions.
>
> **W3 — NavDP pre-training dependence.**
>
> While a good pre-trained model does help, our from-scratch variant achieves SR 48.3 (Table 2), outperforming NavDP (42.2) with only **8$\times$4090$\times$2 days** vs. NavDP's 32$\times$A100$\times$32 days, demonstrating that the large-scale online IL paradigm itself provides even greater value (a detailed compute breakdown is provided in our response to Reviewer oKdX W3/Q1).
>
> **Q1 — Catastrophic forgetting; why no replay buffer?**
>
> Throughout training, benchmark metrics **improve consistently** with no catastrophic forgetting. This stability benefits from 256 environments $\times$ 50 scenes providing high data diversity.
>
> NavOL focuses on **real-time correction and guidance** of the robot's current behavior rather than replaying past experience. Using only current-rollout data with fixed compute per iteration, rollout quality improves monotonically, making the latest data the most informative source while also avoiding dataset bloat.
>
> **Q3 — Inference latency on Go2.**
>
> On the Go2's Jetson Orin, the policy runs at 4.75 Hz, meeting real-time needs. No inference optimizations (e.g., TensorRT, distillation) have been applied yet, a promising future direction.
>
> **Q4 — Goal-agnostic critic and goal-directed behavior.**
>
> All candidates come from the **goal-conditioned diffusion policy**, inherently goal-directed. The critic solely selects the safest among these without introducing off-target alternatives. The goal-agnostic design enables universal obstacle-avoidance scoring based on path-environment interactions, yielding stronger cross-goal generalization.
>
> **Limitations**
>
> We thank the reviewer for these careful observations and valuable concerns. We would like to highlight a few points in response.
>
> **(1) NavDP pre-training dependence.** Please see W3 above.
>
> **(2) Static indoor scenes.** As discussed in W2/Q2, our policy can handle dynamic scenarios in practice, effectively avoiding moving pedestrians during real-world deployment thanks to its RGB-D input. Nonetheless, systematic evaluation under complex and rapidly changing dynamic environments remains a valuable direction for future research.
>
> **(3) Asset pipeline.** We agree that the BlenderProc/IsaacSim pipeline reduces accessibility and should be acknowledged as a limitation. This overhead is limited to one-time training data preparation and does not recur at deployment, since the trained policy deploys **zero-shot** with no map. We will clarify this in the revised version.
>
> **(4) Compute cost.** We agree that requiring 8 RTX 4090s for 2 days remains a meaningful hardware barrier and should be acknowledged as a limitation. Although NavOL requires substantially less compute than NavDP (see W3), we will make this point explicit in the revised version.
>
> We sincerely appreciate these suggestions. A formal limitations discussion will be included in the revised manuscript, and the directions identified above will guide continued research.

---

> > ### Author Rebuttal · Reviewer_LD49 · 2026-04-03
> >
> > My concern has been well addressed and I've updated the score. The author should keep revising the manuscript for better clearance.

---

> > > ### Author Response · Authors · 2026-04-04
> > >
> > > We sincerely thank reviewer for the positive reassessment and for raising the score. We will revise the manuscript to include: a dedicated Limitations section on pre-training dependence, static scenes, asset pipeline accessibility, and hardware cost; improved intuitive explanation of the safety score (Eq. 4); and scene construction cost data. Clarifications on catastrophic forgetting, inference latency, and goal-agnostic critic will also be incorporated.

---

### Official Review · Reviewer_tGTP · 2026-03-13

**Soundness:** 4
**Presentation:** 3
**Significance:** 3
**Originality:** 3
**Overall Recommendation:** 5
**Confidence:** 4

**Summary:**

This paper presents a method for robot navigation in indoor environments. The authors propose an approach to train a navigation policy using online imitation learning to address the distribution shift and compounding errors that typically occur in offline methods. The training alternates between two phases: a rollout phase and an update phase. During the rollout phase, the learned policy is executed in the simulator. Simultaneously, an expert planner generates optimal trajectories to the goal from each state the policy visits. During the update phase, the optimal trajectories from the expert planner are used to finetune the policy. This differs from offline methods that train on fixed datasets. The authors evaluate their method on both simulation and real-world environments. For simulation, they evaluate on the NavDP benchmark and also introduce a new benchmark. They also design and run real-world experiments. In both cases, they show significant improvements over baselines.

**Compliance With Llm Reviewing Policy:**

Affirmed.

**Final Justification:**

The online imitation learning approach proposed by the authors significantly improves navigation performance over baselines, both on simulation and real-world experiments. The authors have answered my questions during the rebuttal and included additional ablations for some method hyper-parameters. I think this paper deserves to be published and I'm increasing my score from 4 to 5 after the rebuttal.

**Key Questions For Authors:**

1. The rollout phase should be explained in more detail. How many waypoints are executed per step? How many times is the policy called per rollout?
2. How is K (the number of steps/waypoints) determined? How does it impact performance?
3. What is the value of F? How does it impact performance?
4. How is the goal provided? Is it an image or is it a position / coordinates? Is it consistent across benchmarks?

**Limitations:**

Yes.

**Strengths And Weaknesses:**

**Strengths**
- Extensive experimental evaluation on simulation benchmarks.
- Comparison against multiple baselines and significant improvements.
- Successful experiments on real environments, which proves cross-domain generalization of the introduced method, as training is done on a Dingo robot in simulation, while real-world experiments are done on a Unitree Go2.
- Introduction of a new benchmark for future work.

**Weaknesses**
- Except for NavDP, the comparisons to baselines may not be entirely fair because the evaluation benchmarks are closer to this work’s training data (scenes built from the 3D-Front dataset) than to the other baselines’ training data (Gibson-2, Matterport3D).
- Comparison to baselines is limited to zero-shot baselines. It would be nice to fine-tune some of them on 3D-Front scenes.
- Some important hyperparameters lack ablations and justification (e.g, F,  K, rollout length per iteration, expert action probability \rho), and some important details are lacking from the experiments (how is the goal specified?).

**Comments / Questions**
5. What does it mean that waypoints are converted to velocity commands via MPC?
6. Why is the goal instruction optional? What does the generator predict if no goal is provided?
7. How long are trajectories on average(in m)?
8. What are the unfavorable initial conditions that make the agents immediately fall or hit obstacles?
9. Table 1. Is there any reason why the performance on Intern-Home is worse than NavDP’s?
10. It would be nice to have an explanation of why some baselines fail more on certain scenes than others (are those scenes more complex?).

**Typos and minor comments**
- The title of Sec.4 is missing an “s” → Experiments.
- In Sec.4.4., in “Train from scratch”: “we try initialize” is missing “to”.
- Fig 2. It would be nice to have f, f’ and V appear in the figure.

---

> ### Author Rebuttal · Authors · 2026-03-31
>
> We sincerely appreciate Reviewer tGTP's positive assessment and thoughtful questions. We respond below.
>
> **W1 & W2 — Comparison fairness and baseline fine-tuning.**
>
> It is worth highlighting that (1) **we never train on any test scene**: all 8 evaluation scenes are held out. Even on NavDP's own benchmark (where NavDP *was* trained), NavOL outperforms NavDP (Table 1), confirming no data-domain bias. (2) All methods are evaluated **zero-shot**: DD-PPO, iPlanner, ViPlanner train on Gibson-2/Matterport3D; NavDP and NavOL on disjoint 3D-Front scenes, following standard navigation protocol.
>
> We acknowledge that retraining baselines would strengthen the comparison; unfortunately reproducing each baseline's pipeline is non-trivial (e.g. NavDP's offline trajectories and rendering). It is worth noting that Table 2 shows NavDP uses **32$\times$A100$\times$32 days**; NavOL from scratch uses only **8$\times$4090$\times$2 days** yet outperforms NavDP (mSR 48.3 vs. 42.2), demonstrating greater efficiency (see Reviewer oKdX W3).
>
> **W3 — Ablations on F, rollout length, $\rho$.**
>
> *$\rho$*: We ablate $\rho \in \{0.0, 0.5, 0.8, 1.0\}$. $\rho$=1.0 (pure policy rollout: SR 43.9) performs worst; $\rho$=0.0 (SR 63.8) is suboptimal as the policy never encounters its own failure modes; $\rho$=0.8 (SR 69.0) best balances policy exploration and expert stabilization (see also Reviewer oKdX Q2).
>
> *F (minimum keypoints per episode, current F=5)*:
>
> | F | mSR | mSPL |
> |---|-----|------|
> | 2 | 67.7 | 61.8 |
> | 5 | 69.0 | 63.7 |
> | 8 | 69.8 | 63.8 |
>
> The performance differences across tested $F$ values are small, indicating that the method is not particularly sensitive to $F$. We keep $F$=5 as the default because it filters trivial episodes while maintaining near-best performance.
>
> *K (K=24)*: Inherited from NavDP's DDPM design; varying K requires backbone redesign.
>
> *Rollout length*: Table 2 shows 8 steps (SR 67.9) and 32 steps (SR 68.5) perform close to the full setting 128 steps (SR 69.0), confirming low sensitivity.
>
> **Q1 — Rollout details.**
>
> The model predicts 24 waypoints. We then execute only a single control action, namely the linear and angular velocities computed by the MPC controller from the waypoints. Each training iteration collects 128 rollout steps across 256 parallel environments (Sec. 4.1).
>
> **Q2 / Q3 — How are K and F determined?**
>
> K=24 is from NavDP. F=5 was empirically set to filter trivial episodes. See W3.
>
> **Q4 — Goal specification.**
>
> All evaluations use point-goal: a 2D relative displacement ($\Delta x$, $\Delta y$) in the robot frame. Start-goal pairs are **randomly sampled** and verified via NavMesh (Sec. 3.2).
>
> **Q5 — Waypoint-to-velocity via MPC.**
>
> We implement a nonlinear MPC based on CasADi following NavDP, modeling the robot as a differential-drive platform. The controller solves for $(v, \omega)$ under constraints ($v_{\max}$=0.5 m/s, $\omega_{\max}$=0.5 rad/s), minimizing tracking error with acceleration penalties.
>
> **Q6 — Why is the goal optional?**
>
> As described in our method (Sec. 3.1), the architecture accepts an optional goal. In goal-conditioned mode the policy navigates toward the target; in goal-free mode it performs exploratory roaming for navigation and collision avoidance. All evaluations use goal-conditioned mode.
>
> **Q7 — Average trajectory length.**
>
> Episodes average over 400 steps at 0.15 m/s ($\sim$30 m).
>
> **Q8 — Unfavorable initial configs in NavDP benchmark.**
>
> Some NavDP benchmark episodes place the agent inside walls or above obstacles at initialization, causing immediate failure. These affect all methods equally, stemming from benchmark artifacts rather than navigation capability.
>
> **Q9 — InternHome performance lower than NavDP.**
>
> Thank you for this careful observation. We attribute the gap to two factors: (1) InternHome scenes are predominantly **long corridors with sparse landmarks**; NavDP's offline data covers such structures thoroughly with performance near its ceiling, leaving limited room for improvement. (2) Some failures are caused by the **Dingo chassis getting stuck** at corridor junctions or thresholds, after which the robot cannot self-recover, and such cases count as failures even though the predicted waypoints are geometrically reasonable. These explain the slightly lower scores without reflecting policy quality.
>
> **Q10 — Baseline variance across scenes.**
>
> Our benchmark spans diverse layouts. Scene 8 has regular corridors matching NavDP's offline data (NavDP 87, Ours 55). Scene 1 has dense furniture, challenging all methods ($\leq$54). Scene 3 is moderate (iPlanner 10, ViPlanner 23, NavDP 37, Ours 80); Scenes 4\~5 are open multi-room layouts favoring point-cloud planners (iPlanner 24\~27, ViPlanner 42\~43) while NavDP struggles (25\~37). NavOL remains strong across all (SR 54\~80), showing online learning adapts well.
>
> **Minor fixes**
>
> We appreciate the reviewer's careful proofreading. All noted typos will be corrected in the revision.

---

> > ### Author Rebuttal · Reviewer_tGTP · 2026-04-04
> >
> > Thank you for the rebuttal. I appreciate that you ran the ablations on F, \rho and rollout length, and that you provided method clarifications. It would be good to add the clarifications to the manuscript as well.
> > I will increase my score.

---

> > > ### Author Response · Authors · 2026-04-04
> > >
> > > We sincerely thank reviewer for the positive reassessment and for increasing the score. We confirm that all suggested clarifications, including the ablations on F, $\rho$, and rollout length, as well as method details on rollout execution, MPC, and goal specification, will be incorporated into the revised manuscript.

---

### Official Review · Reviewer_oKdX · 2026-03-13

**Soundness:** 4
**Presentation:** 4
**Significance:** 3
**Originality:** 3
**Overall Recommendation:** 5
**Confidence:** 3

**Summary:**

This paper introduces a new algorithm for learning navigation policies in indoor scenes. Similar to prior works, it trains in simulation, where the true map can be used by an expert planner to compute ground truth paths. In contrast to prior work which uses non-interactive learning methods such as behavior cloning, this paper uses a DAgger style algorithm to better supervise the navigation policy. They report improved performance on existing navigation benchmarks and introduce a new benchmark with higher quality scenarios.

**Compliance With Llm Reviewing Policy:**

Affirmed.

**Final Justification:**

This paper presents a novel approach for learning navigation policies by leveraging DAgger in simulation. The authors addressed my questions and those of the other reviewers in the rebuttal phase. I will keep my original recommendation (accept) and suggest the authors update the final paper based on the reviewers' feedback.

**Key Questions For Authors:**

1. How does NavOL compare to NavDP when using equivalent amount of compute (i.e. NavDP will use a bigger non-interactive dataset and NavOL will use a smaller initial dataset and then run DAgger).

2. Can you report an ablation experiment on $\rho$ (the probability of picking the learner policy's action) to demonstrate why using $\rho = 0.8$ is beneficial?

**Limitations:**

Yes

**Strengths And Weaknesses:**

Strengths:

This paper introduces a new algorithm for learned navigation, using a DAgger-style approach to more efficiently learn from the privileged expert planner. They demonstrate strong empirical results, improving on existing techniques. The authors also provide numerous qualitative examples of their navigation policy in comparison to prior approaches. The methodology is explained clearly.

Weaknesses:

This approach relies on simulation with a ground truth map to compute expert navigation paths. Building such maps for new locations can be costly.

The paper mentions that avoiding RL helps avoid reward engineering, but then engineer a cost function for the critic to learn.

As the proposed NavOL approach initializes from the pre-trained NavDP policy, the comparison between NavOL and NavDP are comparing policies trained with different amounts of data and compute. It would be good to see an ablation between the two approaches matching compute costs during training.

Minor Issues:

There are typos in Figure 1 (e.g. "Tajectory").
The phrase "We introduce NavOL" is used in two subsequent paragraphs in Section 1. I'm not sure if that was intentional or not.

---

> ### Author Rebuttal · Authors · 2026-03-31
>
> We sincerely thank the reviewer for the positive assessment and recognition of our work. We address each concern below.
>
> **W1 — "Building such maps for new locations can be costly."**
>
> It is worth noting that the trained NavOL policy deploys **zero-shot** to new scenes with no map required. NavMesh is used only during training in simulation to provide path labels for the expert planner. Even for training scenes, NavMesh computation takes only seconds per scene, and the 3D asset pipeline (BlenderProc re-texturing + USD conversion) is a one-time process. Map construction cost is therefore not a deployment bottleneck.
>
> **W2 — The critic requires a cost function, seemingly contradicting "no reward engineering."**
>
> We appreciate this precise observation. It is important to note that the critic shares the ViT/DiT backbone with the actor and both are trained jointly via $\mathcal{L} = \mathcal{L}^{\text{actor}} + \lambda\, \mathcal{L}^{\text{critic}}$. However, $\mathcal{L}^{\text{critic}}$ is **not** a reward signal, it is a **geometric distance supervision**: we compute each waypoint's distance to its nearest obstacle via NavMesh and use this as a deterministic label, requiring only two fixed hyperparameters ($d_{\text{safe}}$, $\alpha$) with no environment-specific tuning. **This is fundamentally different from RL reward engineering**: in RL, meticulous reward design is critical to training stability, a poorly designed reward function can directly destabilize training and degrade policy performance. In contrast, Table 2 demonstrates that our critic is an **auxiliary enhancement** rather than a stability-critical component: removing it ("w/o critic") only decreases SR from 69.0 to 67.1 (1.9-point drop), with no impact on training stability. At inference time, the critic serves as a **passive trajectory ranker**, selecting the safest among multiple diffusion-sampled candidates. The policy's core navigation capability comes from $\mathcal{L}^{\text{actor}}$, which provides **online expert supervision** on the policy's own visited states.
>
> **W3 / Q1 — NavOL initializes from NavDP; is the comparison compute-fair?**
>
> An excellent question. Even from a compute perspective, NavOL is far more efficient than NavDP's offline training:
>
> **Direct compute comparison.** NavDP trains on **32×A100 for 32 days** for offline data collection and training. NavOL's "from scratch" variant (Table 2 ablation) uses only **8×RTX 4090 for under 2 days**, with far less data and compute than NavDP, yet already outperforms it (from-scratch mSR 48.3 vs. NavDP 42.2). This demonstrates that **online interactive supervision is far more data-efficient than offline data accumulation**, expert supervision obtained on the policy's own visited distribution is more effective than offline rendering of massive trajectory datasets.
>
> The full NavOL with NavDP pre-training initialization (additional 16 GPU-days of online training) further improves mSR to 69.0, a +26.8 absolute improvement on top of NavDP's $\sim$1024 GPU-days with only $\sim$1.5% additional compute.
>
> | Method | mSR | mSPL |
> |--------|-----|------|
> | NavDP (original, 32×A100×32 days) | 42.2 | 37.7 |
> | NavOL from scratch (8×4090×2 days) | 48.3 | 45.5 |
> | NavOL (NavDP init + 8×4090×2 days) | **69.0** | **63.7** |
> > NavDP GPU-day estimates are based on its paper.
>
> **Q2 — Please report an ablation on $\rho$.**
>
> Table 2 already includes $\rho$=1.0 ("w/o expert rollout": SR 43.9 vs. full model SR 69.0), showing a large drop when expert guidance is fully removed. We further do ablation studies on $\rho$:
>
> | $\rho$ | mSR | mSPL |
> |---|-----|------|
> | 0.0 | 63.8 | 57.9 |
> | 0.5 | 63.1 | 56.5 |
> | **0.8** | **69.0** | **63.7** |
> | 1.0 | 43.9 | 41.4 |
>
> The result is consistent with expected pattern in DAgger theory: $\rho$=1.0 worst; $\rho$=0.0 suboptimal (the policy never sees its own failure modes); $\rho$=0.8 optimal, balancing policy exploration with expert stabilization.
>
> **Minor fixes**
>
> We appreciate the reviewer's careful proofreading. The typo "Tajectory" in Figure 1 will be corrected, and the repeated introductory phrasing "We introduce NavOL" in Section 1 will be revised for better flow.

---

> > ### Author Rebuttal · Reviewer_oKdX · 2026-04-03
> >
> > Thank you for the rebuttal. I'll order follow-up comments to match how you structured the rebuttal.
> >
> > W1: I agree with everything you wrote. The limitation I was trying to point to is that creating new training maps is a potentially expensive process (I don't see details in the paper on how expensive) and that could be a limitation to scaling this algorithm to handle a more diverse set of environments.
> >
> > W3/Q1: Thank you for the table. I think something like this table would be helpful to include in the paper.
> >
> > Q2: With $\rho = 0$, the expert action is always taken, which I believe would mean the algorithm reduces to behavior cloning.  Looking at the theory in the original DAgger paper (Ross et al 2011), they use $\beta$ where $\beta = 1$ means always use the expert (i.e. their $\beta$ is the opposite of this paper's $\rho$: $\beta = 1 - \rho$). DAgger theory requires that $\frac{1}{N} \sum_{i=1}^N \beta_i \to 0$ as $N \to \infty$, which I believe would mean $\rho$ getting close to 1. I'm therefore not clear why you say "$\rho=1.0$ worst" based on the original DAgger theory. Given this confusion, I don't know what to make of the ablation results shown in the rebuttal.

---

> > > ### Author Response · Authors · 2026-04-04
> > >
> > > We sincerely thank the reviewer for the thoughtful follow-up questions and the valuable suggestion to include the compute comparison table. We address each point below.
> > >
> > > **W1 — Scene construction cost as a scaling limitation.**
> > >
> > > Creating training maps involves two parts: **(1) raw mesh acquisition** and **(2) post-processing**.
> > >
> > > In our current setup, we directly use meshes from the 3D-Front dataset. The post-processing (normal correction, height normalization, GLB-to-USD conversion, NavMesh generation) takes approximately 1 minute per scene and is a one-time cost.
> > >
> > > For scaling beyond existing datasets, we have also explored a scene reconstruction pipeline from real-world videos: COLMAP for camera pose extraction, 2D/3D Gaussian Splatting (with geometric constraints for higher-quality geometry), followed by TSDF-based mesh extraction. This pipeline takes approximately 1 hour per scene on a single RTX 4090, meaning hundreds of real-world scenes can be processed within a few days on 8 GPUs. We will discuss this scaling path and acknowledge the pipeline cost as a limitation in the revised manuscript.
> > >
> > > **W3/Q1 — Compute comparison table.**
> > >
> > > Thank you for the suggestion. We will include the compute comparison table (NavDP vs. NavOL from scratch vs. full NavOL) in the revised manuscript.
> > >
> > > **Q2 — Relationship between $\rho$ and DAgger theory.**
> > >
> > > As the reviewer correctly identified, our $\rho$ corresponds to $1 - \beta$ in DAgger, where $\beta$ is the probability of executing the expert action. In standard DAgger, $\beta$ gradually decreases over iterations so that the policy progressively takes over. NavOL instead uses a **fixed** $\rho = 0.8$ ($\beta = 0.2$) throughout all training iterations.
> > >
> > > Why $\rho = 1.0$ ($\beta = 0$, no expert execution) performs worst: without any expert involvement during rollouts, the untrained policy controls all execution and frequently enters catastrophic states (collisions, getting stuck). Although the expert still provides action labels for supervision, the visited states are dominated by failure modes, and the policy struggles to learn useful behavior from such degraded rollouts. The original DAgger paper reports a similar finding in their Mario experiment: without expert execution the agent repeatedly gets stuck, whereas "using the expert a small fraction of the time ... unstucks marios and makes it collect a wider variety of useful data." Our fixed $\beta = 0.2$ serves exactly this role, keeping rollouts informative throughout training.

---

### Decision · Program_Chairs · 2026-04-30

**Decision:**

Accept (regular)

**Comment:**

This paper employs DAgger-style imitation learning using a global planner to provide expert demonstations to refine a pretrained navigation diffusion policy. Getting this system to work is an impressive engineering feat, with experimental results demonstrating the advantages of the approach. Originality is a weak point: the paper is combining together a lot of existing components without creating any specific new theory. The combination of these ideas (along with a safety critic) is interesting, however as one reviewer states: "The work is more of a system-level innovation and an optimization of the learning paradigm rather than a fundamental change to fundamental algorithms." This leaves some questions of whether ICML is the appropriate venue for this work rather than a more systems-based robotics conference with which the contributions might more appropriately align.

Given the enthusiasm of the reviewers, I recommend acceptance with the caveat about the fit of the paper for ICML vs. a robotics venue.